# Multimodal Generative Composition Recommendation

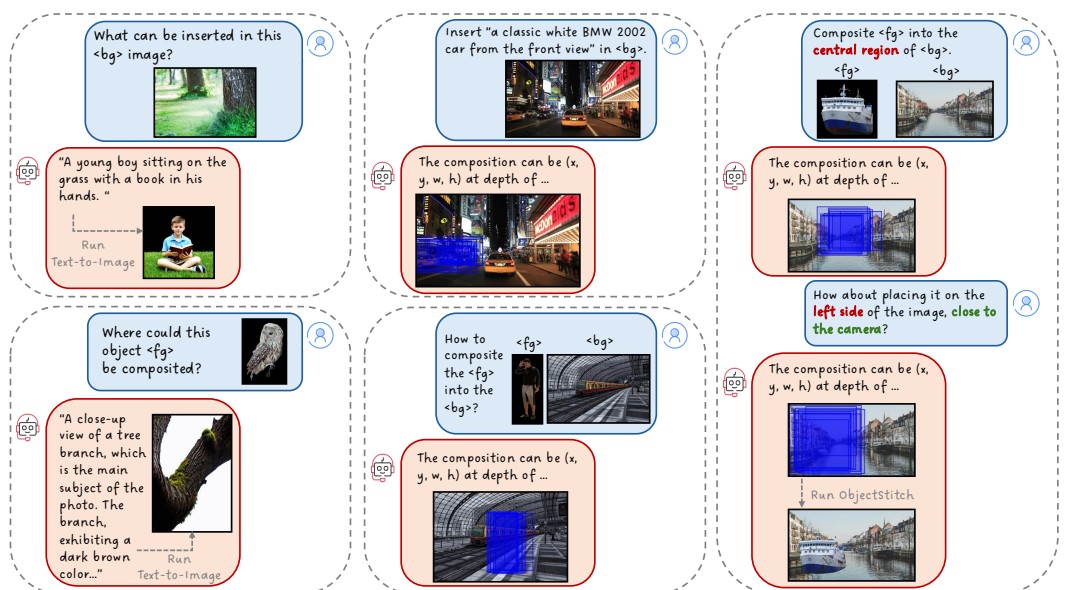

Figure 1: Various use cases for our model. **Left**: Generating compatible foreground suggestions for a given background or vice versa. **Middle**: Predicting plausible object placements based on user-provided foreground input—either in text or image form—within a specified background. **Right**: Offering user-controllable suggestions for object placement.

## Abstract

Compositing an object into a given image is a common task in image editing, requiring both creative ideation and technical precision to achieve harmony. Professionals start by brainstorming concepts, then scale and position elements to integrate seamlessly within the image. While recent diffusion models have made significant progress in pixel harmonization, suggesting suitable concepts as well as recommending compatible composition locations and scaling remains a less explored and challenging task. In this work, we leverage the advanced reasoning capabilities of Multimodal Large Language Models (MLLMs) to address these challenges. We first propose a data pipeline that automatically generates diverse, high-quality, large-scale training data from an internet-scale stock image. Using this dataset, we fine-tune MLLMs with enhanced projector designs and targeted data augmentation to achieve robust content recommendation and precise object placement, demonstrating strong performance against prior methods. Our model supports flexible input options—either image or text—alongside user-defined placement control, offering designers a new level of creative flexibility. Finally, we showcase the model's impact in real-world editing workflows where our model achieves state-of-the-art performance consistently on image composition benchmarks, including our self-created in-the-wild evaluation dataset Composition1K. The code and the Composition1K dataset are provided at
https://anonymous.4open.science/r/MGCR.

# 1 INTRODUCTION

Image composition—deciding what to add and where to place it—is fundamental to image editing and synthesis. Professional designers routinely curate backgrounds and foregrounds and adapt them into coherent scenes that meet aesthetic and physical plausibility criteria. Yet, despite rapid progress in generative imaging (Lin et al., 2018; Niu et al., 2022; Tan et al., 2018; Bomatter et al., 2021; Zhu et al., 2023b; Tarrés et al., 2024), current tools provide little support for recommending elements that are semantically and spatially compatible with a given image, nor do they offer designer-friendly guidance. As a result, finding complementary content, resizing and placing objects, and achieving spatial harmony remain time-consuming and heavily reliant on expert intuition.

This paper addresses two main challenges in image composition: conditional content recommendation and object placement. Unlike the tasks typically addressed in visual question answering, content recommendation and object placement require a deep understanding of spatial relationships and semantic context, and sometimes even complex spatial reasoning ability. These challenges are significant and meaningful tasks in their own right. For content recommendation, we examine whether our model can generate prompts for precise, aesthetically coherent foreground elements for a given background and, conversely, suggest plausible backgrounds for a provided foreground. For object placement, we investigate how our model can offer robust, contextually appropriate, and user-controllable suggestions for positioning, scale, and depth of foreground elements within a background. While existing approaches to object placement are widely studied in both generative (Lin et al., 2018; Tripathi et al., 2019; Zhang et al., 2020; Zhou et al., 2022; Zhang et al., 2023; Yun et al., 2024) and discriminative models (Liu et al., 2021; Niu et al., 2022; Zhu et al., 2023b), we advance these efforts by leveraging the reasoning capabilities of Multimodal Large Language Models (MLLMs). By fine-tuning MLLMs on carefully curated datasets, we achieve enhanced effectiveness, robustness, and control, allowing our model to provide detailed, contextually appropriate recommendations that adhere to fundamental compositional logic. Moreover, the MLLM-based automatic composition recommendations can significantly reduce the workload on users, facilitating more efficient and intuitive image editing and design.

To realize these objectives, we build on LLaVA (Liu et al., 2024c;a) and a scalable data pipeline that integrates filtering, segmentation, local region captioning, and layer-wise object removal of stock imagery. We then perform instruction tuning to align the model with the two tasks. Our contributions are summarized as follows:

- We propose a scalable and effective data generation pipeline to improve compositional content recommendation and object placement. This pipeline integrates image processing steps such as filtering, segmentation, captioning, and object removal to produce high-quality, dense data, enhancing learning efficiency and generalization.

- We design an MLLM architecture with independent projectors to effectively extract and fuse foreground images, background images, and text, achieving superior integration over the baseline. Also, we implement targeted data augmentation to further boost performance.

- Through MLLM fine-tuning, our model can suggest placements for flexible foreground modalities (image or text) and enables user control via language input.

- For thorough evaluation, we construct 1149 foreground-background pairs from in-the-wild data, dubbed as Composition1K. Our model demonstrates top object placement performance, confirmed by standard metrics on datasets like OPA (Liu et al., 2021) and user studies on our Composition1K dataset.

# 2 RELATED WORK

**Image Content Recommendation** Recent studies have advanced the field of content-aware object generation recommendation (Qiu et al., 2020; Shukla et al., 2023; Chiu et al., 2024; Fanelli et al., 2025). For example, Shukla et al. (2023) utilized a scene graph to detect absent objects and their interrelations within existing scene elements. However, these methods rely on background images with predefined missing areas, limiting flexibility in object type and placement. Our research focuses on recommending semantically compatible objects for clean, intact backgrounds and suggesting background content for given foreground objects, enabling more flexible applications.

**Scene Graph- and Layout-based Composition** Scene graphs and layout-conditioned generation provide structured control for compositional synthesis and editing (Zhang et al., 2024; Farshad et al., 2023; Gao et al., 2024; Feng et al., 2024; Wang et al., 2024). Different from these controllable editing/generation pipelines that require a concrete textual prompt, our system focuses on automatic recommendation: given a clean background/foreground(s), it suggests semantically compatible foreground(s)/background; moreover, it outputs accurate, fine-grained placement proposals. This serves non-expert users who lack concrete ideas by recommending what to add and where.

**Object Placement and Compositional Object Insertion** Deep learning has advanced object placement through diverse approaches. PlaceNet (Zhang et al., 2020) leveraged self-supervised learning to place objects contextually, while GracoNet (Zhou et al., 2022) framed placement as a graph completion task. TopNet (Zhu et al., 2023b) applied transformers to model foreground-background correlations and predict placement heatmap. In parallel, pixel-generative composition methods have also progressed, focusing on identity-preserving object insertion. ObjectStitch (Song et al., 2022) utilized diffusion models, Anydoor (Chen et al., 2024a) supported multi-object placement, and Tarrés et al. (2024) eliminated mask constraints with realistic shadows and reflections. Our work addresses the initial stage of object placement recommendation, which leverages generative models to enable user-friendly image composition.

**Multimodal LLMs** Recent advancements in computer vision have been propelled by the development of MLLMs (Liu et al., 2024c; Zhu et al., 2023a; Chen et al., 2024c; Yang et al., 2024a). A notable series in this development is the LLaVA series (Liu et al., 2024c;a;b), setting a precedent for visual instruction tuning. The paradigm has been extended to many areas including image grounding (Zhang et al., 2025), video understanding (Lin et al., 2023; Li et al., 2024a) and medical image understanding (Li et al., 2024b). However, the application of these models in image composition tasks remains under-explored. We aims to fill this gap by focusing on the application-oriented aspects of image composition. We leverage the generalization and reasoning capabilities of large vision-language models to enhance the recommendation process, providing substantial support and improvement to the image composition workflow.

# 3 METHOD

## 3.1 SCALABLE DATA GENERATION PIPELINE

Our scalable data generation pipeline comprises MLLM-based data filtering, inpainting mask merging, object removal, and full training bundle formation.

**MLLM-based Data Filtering** Stock image databases often contain unsuitable data (e.g., blank backgrounds, repetitive objects) that degrade training. We use InternVL 1.5 (Chen et al., 2024c) to filter out such images (see Fig. 2), retaining $\sim 70\%$ of the data. This yields 709K background images and 1.3M background-foreground pairs.

**Inpainting Mask Merging** Using pre-computed segmentation masks (Qi et al., 2022a), we merge masks by: 1) selecting one random mask; 2) selecting a random subset; or 3) selecting all masks. This varies object density and prevents the model from overfitting to inpainting artifacts by removing multiple objects.

**Object and Effect Removal from Background** State-of-the-art inpainting models often struggle with tight masks, shadow removal, and residual artifacts. To address this, we adopt an internal version of ObjectDrop (Winter et al., 2024) for layer-wise object removal, ensuring high-quality background generation without leftover cues that cause overfitting.

**Automatic Training Bundle Generation** We construct training bundles for content recommendation (predicting fg/bg captions) and controllable placement (predicting bounding boxes and depth). We use ViP-LLaVA (Cai et al., 2024) for fg captions, InternVL 1.5 (Chen et al., 2024c) for bg captions, Depth Anything V2 (Yang et al., 2024b) for depth maps, and GPT-4 for varied instructions. See Appendix G for details.

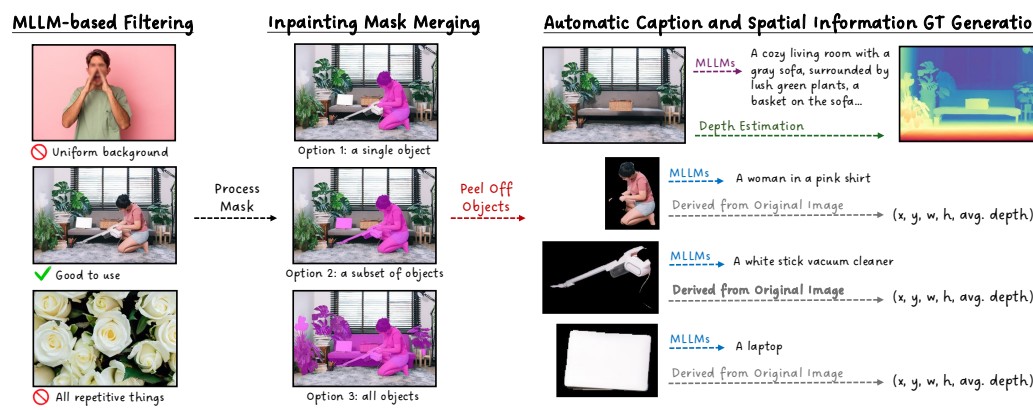

Figure 2: Overview of our data preprocessing pipeline. Starting with an internet-scale image database, we first apply off-the-shelf MLLM (Chen et al., 2024c) to filter out images irrelevant to our task. Next, we generate entity segmentation (Qi et al., 2022b) masks and apply inpainting in one of the three randomly chosen modes: targeting a single object, a subset of objects, or all objects. We then separate the selected objects from the original image to create paired 'clean' background images and isolated foreground objects, using our customized variant of ObjectDrop (Winter et al., 2024). Finally, our pipeline automatically generates dense captions for the 'clean' background using InternVL 2.0 (Chen et al., 2024c), captions each regional foreground object with ViP-LLaVA (Cai et al., 2024), and computes depth map with Depth Anything V2 (Yang et al., 2024b).

## 3.2 DATA AUGMENTATION TO OVERCOME OVERFITTING

After data generation through the pipeline in Section 3.1, we obtain pairs of fg images and "clean" bg images, where the corresponding fg has been removed. However, directly using these pairs for training can lead to overfitting in object placement tasks for two main reasons. First, the paired fg and bg from the same original image are perfectly harmonized in terms of lighting, contrast, and resolution, allowing the model to exploit these low-level matching cues to predict placement. Second, residual inpainting artifacts may still be present, which the model could learn as shortcuts.

To address these issues, we apply different random transformations to the color, brightness, and resolution of both fg and bg images, disrupting these cues to prevent shortcut learning. Additionally, we randomly inpaint multiple regions of the bg to introduce high-frequency artifacts across the image, further reducing the risk of overfitting to specific artifacts. These techniques encourage the model to focus on meaningful spatial and semantic relationships rather than relying on superficial visual cues. The ablation study also confirms the importance of these augmentations in Sec. 4. The detailed implementation of data augmentation is in Appendix E.

## 3.3 MODEL TRAINING AND DESIGN

### 3.3.1 COMPOSITIONAL CONTENT RECOMMENDATION

For visual content recommendation, we target two scenarios: suggesting a fg element given a bg, and suggesting a bg given a fg. The detailed illustration is shown at the top of Fig. 3.

Given a bg image, we use an MLLM to understand the context and recommend compatible fg objects. First, we encode the bg image using CLIP (Radford et al., 2021) and project it into the MLLM's embedding space. We then prompt the MLLM with questions like "What would be a compatible foreground concept?" to generate suitable object descriptions. This setup naturally supports multi-round interactions, allowing the model to recommend multiple fg objects. For additional suggestions, we simply continue with prompts like "What else could be a compatible foreground concept?" Each prompt includes the bg image and past responses to inform the model's recommendations. The conversation ends when suitable suggestions are exhausted or become repetitive.

Similarly, for bg scene recommendations given one or more fg objects, we encode the fgs and prompt the MLLM with questions such as "What background would be compatible with this set of objects?" The MLLM is trained to predict a detailed bg description. We found that a single MLLM can learn

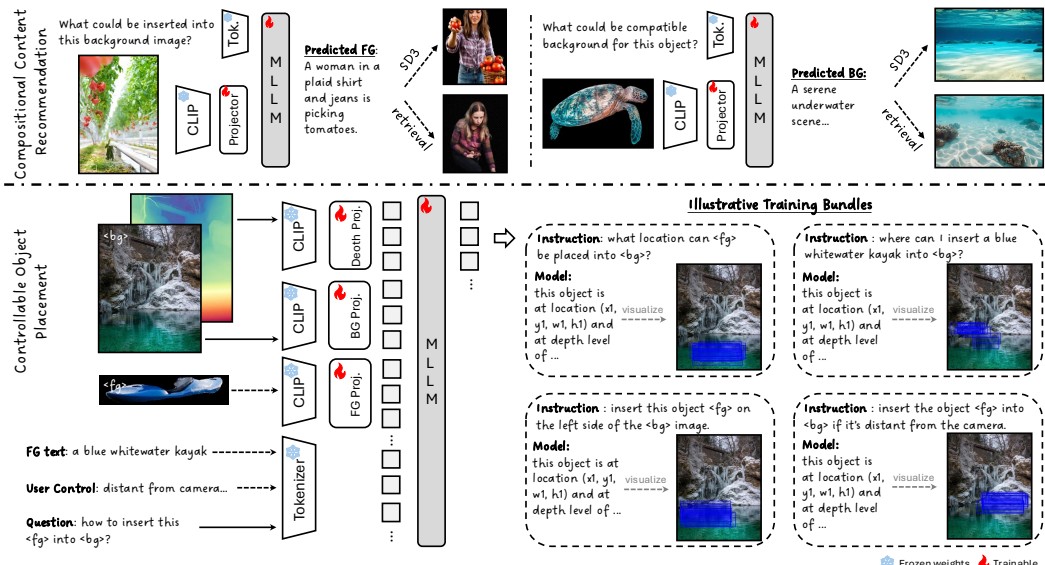

Figure 3: **Top**: Our model suggests compatible foreground elements when users provide a background image and a question, or vice versa. **Bottom**: On the left, we show an overview of our model's inputs and outputs for controllable object placement training, where inputs can optionally include a foreground as text or image and user controls, such as 'distant from camera,' indicated by the dashed line. On the right, we illustrate representative training bundles and inference use cases, with blue bounding boxes indicating 10 random samples.

both tasks well enough. Once trained, we use an off-the-shelf text-to-image model, such as Stable Diffusion 3 (Esser et al., 2024), to synthesize images based on the recommendations, allowing users to visualize and select options for subsequent composition.

### 3.3.2 CONTROLLABLE OBJECT PLACEMENT

For object placement, we also train MLLM by using bg, fg (in either text or image format), optional user controls, and instructions as inputs, and generating placement location, scaling, and average depth as outputs, as shown at the bottom of Fig. 3. Below, we summarize several key designs:

**Depth Prediction** Unlike prior object placement methods (Tripathi et al., 2019; Zhang et al., 2020; Zhou et al., 2022; Zhu et al., 2023b; Zhang et al., 2023) that predict only a 2D bounding box, our model also predicts the average depth of the inserted object. This depth information enables positioning of objects behind existing elements in the image by generating occlusion masks through depth comparisons, as shown in the rightmost section of Fig. 7. To enhance depth prediction and spatial understanding, we jointly encode the pre-computed depth map (from Yang et al. (2024b)) with the bg RGB image.

**Separate Projectors** In prior MLLM works (Li et al., 2024c;a) that support multi-image inputs, a shared projector is typically used to transform CLIP features into token space. However, this design is not ideal for our case, where we need to encode a bg image, bg depth map, and fg image, as each requires extracting different types of information. Specifically, there is a significant modality gap: spatial layout information is derived from the bg image and depth map (geometry), while semantic and pose information is critical for the fg image (texture/objectness). Forcing these distinct modalities through a single projector degrades performance as the model struggles to align them effectively. To address this, we propose using three independent projectors to separately transform the bg, bg depth map, and fg features. Denoting the input bg, bg depth, fg images as $I_{\text{bg}}, I_{\text{bd}}, I_{\text{fg}}$, the CLIP vision encoder as $\mathcal{E}$ and the bg projector, bg depth projector and fg projector as $\mathcal{P}_{\text{bg}}, \mathcal{P}_{\text{bd}}$ and $\mathcal{P}_{\text{fg}}$ respectively, we have

$$\mathbf{e}_{\text{bg}} = \mathcal{E}(I_{\text{bg}}), \mathbf{h}_{\text{bg}} = \mathcal{P}_{\text{bg}}(\mathbf{e}_{\text{bg}}); \mathbf{e}_{\text{bd}} = \mathcal{E}(I_{\text{bd}}), \mathbf{h}_{\text{bd}} = \mathcal{P}_{\text{bd}}(\mathbf{e}_{\text{bd}}); \mathbf{e}_{\text{fg}} = \mathcal{E}(I_{\text{fg}}), \mathbf{h}_{\text{fg}} = \mathcal{P}_{\text{fg}}(\mathbf{e}_{\text{fg}});$$

and the MLLM output $Y$ probability function is

$$p_\theta\Big(Y \mid X, I_\text{bg}, I_\text{bd}, I_\text{fg}\Big) = \prod_{t=1}^{T} p_\theta\Big(y_t \mid X, \mathbf{h}_\text{bg}, \mathbf{h}_\text{bd}, \mathbf{h}_\text{fg}, y_{<t}\Big),$$

where $X$ is the textual input (prompt), $y_{<t}$ denotes all previously generated tokens $(y_1, \cdots, y_{t-1})$, $T$ represents the output token length and $\theta$ represents the parameters of the entire MLLM. As shown in Tab. 4, this simple yet effective design significantly improves performance.

**Flexible Input Modality and User Control**   One of the primary reasons for using MLLM in object placement tasks is its natural adaptability to flexible text or image input modality and user control through language, offering a level of creative flexibility unavailable in previous approaches (Tripathi et al., 2019; Zhang et al., 2020; Zhou et al., 2022; Zhu et al., 2023b; Zhang et al., 2023). Leveraging the rich training bundles discussed in Sec. 3.1, we optionally include either a fg image or a fg caption as input, as well as user-control text such as "distant from the camera." This approach allows the model to interpret various user intents and adjust object placement accordingly. Additionally, to evaluate the precision of user control, we constructed a synthetic benchmark of 100 spatial commands (e.g., "top-left", "far"). Our model achieves $> 85\%$ accuracy in following these directives, demonstrating its robustness to diverse phrasing.

While our model architecture builds on the LLaVA framework—similar to many recent VLM approaches—our proposed depth projection fusion, separate visual projections for fg/bg, and carefully designed data augmentations are key to achieving stable and generalized training for this task. These seemingly subtle yet critical design choices offer valuable insights for future work and contribute meaningful novelty within the context of our problem.

## 4 EXPERIMENT

### 4.1 DATASET AND EVALUATION METRIC

**OPA (Liu et al., 2021).**   The most commonly used dataset for the object placement task is Object-Placement-Assessment (OPA) (Liu et al., 2021). The detailed description of OPA is provided in Appendix C. For a fair comparison on OPA, we train our object placement model on a instruction dataset we created to fit our model input using only the positive samples, which constitute approximately one-third of the training set. Following the evaluation protocol in prior literature (Zhou et al., 2022; Zhu et al., 2023b; Tarrés et al., 2024), we evaluate on the same validation set of OPA, which consists of 3,588 images, using the FID and LPIPS (Zhang et al., 2018) score calculated between the positive testing images and copy-and-pasted composites. To further validate our results, we include the recently proposed CMMD score metric (Jayasumana et al., 2024) in the same evaluation set.

**Large Corpus of Stock Images Val Set.**   We sample 10K images from a large corpus of stock images and perform exactly the same process as the training data pipeline to generate bg and fg pairs. The 10K testing set has no overlap with our training distribution and are samples with only one fg selected. Since the bg and fg images are from the same image, we have the ground truth (GT) bounding boxes for evaluation. Following Zhu et al. (2023b), we adopt the classic meanIOU as the evaluation metric. We also include FID and LPIPS score calculated between the original images and copy-and-pasted composites as evaluation metrics. To test the content recommendation performance, we randomly select 1K from these 10K data and perform fg and bg recommendations. We test both the BERT (Kenton & Toutanova, 2019) score of the suggested captions w.r.t. the GT captions and the CLIP score of the generated images w.r.t. the GT images.

**Composition1K Test Set.**   To test the performance of the object placing abilities of the methods in real-world scenarios, we collect a set of 1149 image bg fg pairs (195 scenes) with compatible semantics. For bg images, we select from DIV2K (Agustsson & Timofte, 2017), Flickr (Plummer et al., 2015), and Google Images. For fg object images, we select from OpenImages v7 (Kuznetsova et al., 2020; Benenson et al., 2019). Due to the lack of GT location and human annotations, we conduct a user study, which reflects the human's satisfactory rate of the reasonability, plausibility, and aesthetics of the composited images. Specifically, we collect preference results of 60 volunteers on 300 randomly selected bg fg pairs from the Composition1K dataset. For each example, the users

Table 2: Quantitative Comparison of Methods across Various Metrics on OPA, Stock, and Composition1K Datasets. * denotes performance reported in Zhou et al. (2022), † denotes performance reported in top and ‡ denotes performance reported in Zhang et al. (2023). **On OPA, our approach uses only the positive samples (around $1/3$ of the training data)**. Best performance in **bold**.

| Methods | OPA | | | Stock | | | Composition1K | |
|---|---|---|---|---|---|---|---|---|
| | FID (↓) | CMMD (↓) | LPIPS (↓) | FID (↓) | mIOU (↑) | LPIPS (↓) | Satisfactory (↑) | MLLM (↑) |
| Random Sampling | 31.93 | 0.118 | 0.274 | 11.98 | 0.160 | 0.301 | 0.087 | 46.60 |
| TERSE (2019) | 46.94*† | 0.060 | 0.236 | 6.209 | 0.223 | 0.273 | 0.096 | 18.44 |
| PlaceNet (2020) | 36.69*† | 0.039 | 0.215 | 7.326 | 0.204 | 0.250 | 0.126 | 49.22 |
| GracoNet (2022) | 27.75*† | 0.051 | 0.242 | 5.963 | 0.255 | 0.285 | 0.124 | 37.07 |
| TopNet (2023b) | 23.49† | 0.021 | 0.200 | 14.68 | 0.115 | 0.302 | 0.161 | 55.83 |
| IORPE (2023) | 21.59‡ | - | - | - | - | - | - | - |
| CSENet (2024) | 17.51 | 0.020 | 0.197 | 8.925 | 0.218 | 0.284 | 0.186 | 52.45 |
| BOOTPLACE (2025) | 19.36 | 0.022 | 0.208 | 7.653 | 0.231 | 0.268 | 0.214 | 56.32 |
| LLaVA-1.5-13B (2024a) | 51.53 | 0.317 | 0.383 | 9.588 | 0.265 | 0.325 | 0.149 | 57.79 |
| InternVL-2.0-26B (2024b) | 34.33 | 0.193 | 0.325 | 13.89 | 0.236 | 0.503 | 0.091 | 59.10 |
| GPT-4o with ICL (2024) | 19.94 | 0.019 | 0.193 | 8.725 | 0.244 | 0.243 | 0.298 | 59.97 |
| Qwen2.5-VL (2025) | 24.14 | 0.026 | 0.211 | 10.43 | 0.251 | 0.287 | 0.245 | 58.83 |
| InternVL3.0 (2025) | 28.23 | 0.074 | 0.229 | 12.06 | 0.247 | 0.352 | 0.201 | 59.42 |
| Gemini 2.5 Pro with ICL (2025) | 18.65 | 0.019 | 0.191 | 7.982 | 0.256 | 0.239 | 0.312 | 60.15 |
| Ours | **15.07** | **0.017** | **0.188** | **4.084** | **0.569** | **0.235** | **0.359** | **61.25** |

are asked to choose all the candidates (9 total, one for each method) that they consider satisfactory, and we report the percentage of images users find satisfactory. We also employ claude-3-7-sonnet-20250219 as an MLLM judge to give a score out of 100 to evaluate the composited images using each method. The MLLM scores mainly serve as an objective metric, and the human preferences from the user study are more reliable.

## 4.2 EVALUATING CONTENT RECOMMENDATION

For content recommendation task, we compare our model against SOTA MLLMs, including LLaVA-1.6-13B (Liu et al., 2024c), InternVL-2.0-26B (Chen et al., 2024c), and ChatGPT-4o (OpenAI, 2024). Our model is trained on a custom instruction dataset formatted for caption recommendation using the 709K training data. Tab. 1 shows that our model achieves superior BERT (Kenton & Toutanova, 2019) scores in both fg and bg prediction tasks. Additionally, we generate images using the predicted prompts from each model and compute the CLIP (Radford et al., 2021) score between the generated and original bg/fg. Our model also outperforms others on this metric. The qualitative comparisons are provided in Fig. 8 in Appendix A.

Table 1: Evaluation of Compositional Content Recommendation on the validation set of the large corpus of stock images dataset. In BERT score, P denotes precision, R denotes recall and F1 denotes F-1 score. BG2FG denotes background to foreground generation and FG2BG denotes foreground to background generation.

| Methods | Task | BERT score (P/R/F1) (↑) | CLIP score (↑) |
|---|---|---|---|
| LLaVA-1.6-13B (2024b) | BG2FG | 0.833/0.851/0.842 | 0.640 |
| | FG2BG | 0.853/0.848/0.850 | 0.560 |
| InternVL-2.0-28B (2024c) | BG2FG | 0.860/0.877/0.868 | 0.640 |
| | FG2BG | 0.859/0.845/0.852 | 0.663 |
| GPT-4o (2024) | BG2FG | 0.848/0.873/0.860 | 0.647 |
| | FG2BG | 0.858/0.846/0.852 | 0.664 |
| Ours (specific) | BG2FG | **0.917/0.915/0.916** | **0.678** |
| | FG2BG | **0.884/0.883/0.883** | **0.709** |
| Ours (unified) | BG2FG | 0.914/0.915/0.914 | 0.676 |
| | FG2BG | 0.882/0.881/0.881 | 0.697 |

## 4.3 EVALUATING OBJECT PLACEMENT

For the object placement task, we benchmark against random sampling, specialist object placement methods (Tripathi et al., 2019; Zhang et al., 2020; Zhou et al., 2022; Zhu et al., 2023b; Qin et al., 2024; Zhou et al., 2025), and recent generalist MLLMs (Liu et al., 2024c; Chen et al., 2024c; Bai et al., 2025; Zhu et al., 2025; Comanici et al., 2025). For GPT-4o (OpenAI, 2024) and Gemini-2.5-Pro (Comanici et al., 2025) we adopt 5-shot example selected from our 709k training data for in context learning (ICL) (Brown et al., 2020). For our model, after finetuning on the 709K training dataset, we use random seed 0 to generate bounding box recommendations for all the experiments. As shown in Tab. 2 and Fig. 4, our model consistently outperforms existing meth-

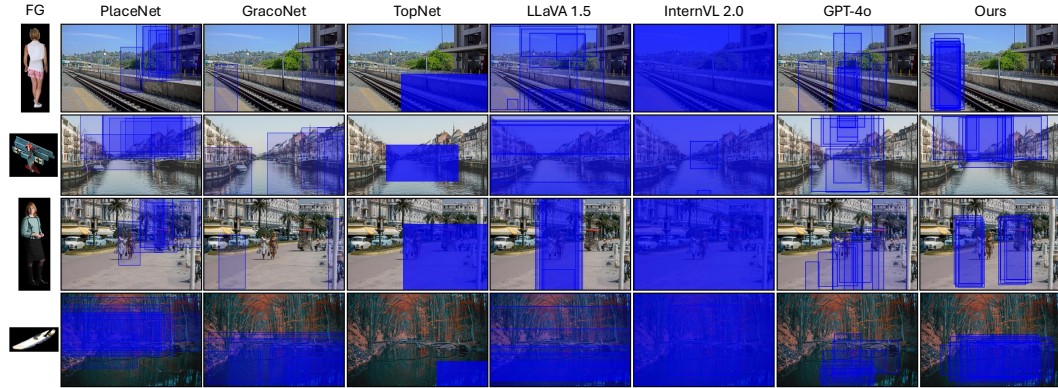

Figure 4: Qualitative comparison. We include three traditional deep models—PlaceNet (Zhang et al., 2020), GracoNet (Zhou et al., 2022), and TopNet (Zhu et al., 2023b)—as well as three MLLMs: LLaVA 1.5 (Liu et al., 2024a), InternVL 2.0 (Chen et al., 2024b), and GPT-4o. For each fg-bg image pair, we sample 10 times for visualizing diverse predictions.

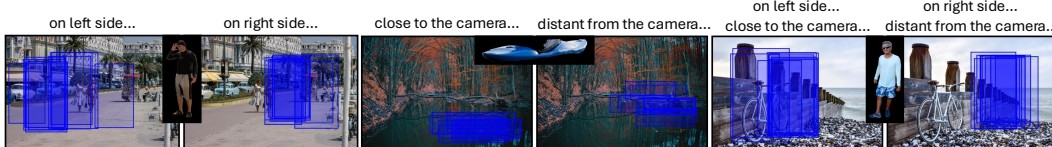

Figure 5: Controllable object composition from our model, with the control text at the top.

ods on both the OPA dataset and the Stock corpus, based on the comprehensive metrics detailed in Sec. 4.1. On the Composition1K test set, both user studies and Claude 3.7 Sonnet further validate the superiority of our approach, with users and Claude 3.7 Sonnet overwhelmingly preferring our model, even slightly over GPT-4o and Gemini-2.5-Pro. The bg and fg pairs in the Composition1K dataset are only compatible in semantics; hence, for most cases, all the methods fail to generate final composited images that look satisfactory to human beings. Nevertheless, our current model still exhibits failure cases such as sometimes overly large bounding box suggestions, as discussed in Appendix P. To further underscore the robustness of our model, we conduct fine-grained performance analyses across diverse testing scenarios, as shown in Fig. 9 in Appendix B. Moreover, the controllable object placement results are illustrated in Fig. 5. More implementation details are in Appendix H and model inference speed is in Appendix Q.

## 4.4 ABLATION STUDY

**Ablation of the data generation pipeline** We ablate the MLLM-based image sample filtering technique and the object removal process in the automated data generation pipeline. As shown in Tab. 3, both practices contribute to the object placement performance of our model. The object removal process is particularly crucial in that if the bg contains objects, the MLLM fails to learn the implicit composition relationship between the bg and fgs. We further study the relationship of our model's object placement accuracy with the training instruction data size. We train all the models for 2 epochs, and we run each experiment 5 times independently. From Fig. 6, we observe a rough

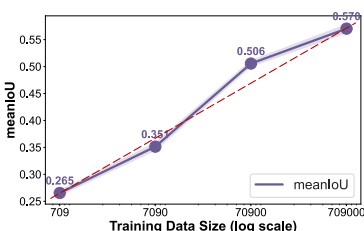

Figure 6: MeanIoU on the Large Corpus of Stock Images Val Set w.r.t. the log-scale training data size. **Scaling law observed**.

scaling law of IoU performance w.r.t. varying training data size on Stock Images Val Set.

**Ablation of the components of proposed framework** As stated in Sec. 3, we proposed targeted data augmentation and the use of separate projectors. We evaluate the effectiveness of these com-

Table 3: Ablation study of MLLM-based Filtering and Object Removal on the Large Corpus of Stock Images Val Set. MF denontes MLLM-based filtering and OR denotes Object Removal.

| MF | OR | FID ($\downarrow$) | meanIOU ($\uparrow$) |
|----|----|------|---------|
| ✗ | ✗ | 9.475 | 0.263 |
| ✗ | ✓ | 6.740 | 0.419 |
| ✓ | ✗ | 9.278 | 0.294 |
| ✓ | ✓ | **4.084** | **0.569** |

Table 4: Ablation study of the different components of our training approach on the Large Corpus of Stock Images Val Set. For each metric, results are shown as specific / unified model. IF denotes Instruction Finetuning, ILF denotes Instruction Location Format, DA denotes Data Augmentation, and SPW denotes Separate Projector Weight.

| baseline | IF | ILF | DA | SPW | FID ($\downarrow$) | meanIOU ($\uparrow$) |
|----------|----|----|----|----|------|---------|
| ✓ | | | | | 9.588 / 9.588 | 0.265 / 0.265 |
| ✓ | ✓ | | | | 5.874 / 5.601 | 0.420 / 0.437 |
| ✓ | ✓ | ✓ | | | 5.725 / 5.429 | 0.448 / 0.455 |
| ✓ | ✓ | ✓ | ✓ | | 5.049 / 4.844 | 0.478 / 0.503 |
| ✓ | ✓ | ✓ | ✓ | ✓ | **4.084 / 3.764** | **0.569 / 0.579** |

Table 5: Ablation study of Depth Input on OPA and Stock datasets.

| Method | OPA FID ↓ | OPA CMMD ↓ | OPA LPIPS ↓ | Stock FID ↓ | Stock mIOU ↑ | Stock LPIPS ↓ |
|--------|-----------|------------|-------------|-------------|--------------|---------------|
| Ours (w/o Depth) | 16.84 | 0.019 | 0.192 | 5.214 | 0.538 | 0.241 |
| **Ours (Full)** | **15.07** | **0.017** | **0.188** | **4.084** | **0.569** | **0.235** |

ponents, along with different bounding box parameterizations. For bounding box parameterization, we hypothesize that predicting the center point $(x, y)$ first, followed by width and height $(w, h)$, aligns better with autoregressive language reasoning compared to the corner coordinates format $(x_1, y_1, x_2, y_2)$ (ILF in Tab. 4). As shown in Tab. 4, our results confirm that each design component indeed contributes to the model's overall effectiveness.

**Ablation of Depth Input** We conduct an additional ablation study to verify the importance of depth input for our model. As shown in Tab. 5, removing the depth input consistently degrades performance across all metrics on both OPA and Stock datasets. Specifically, without depth input, the FID score on OPA increases from 15.07 to 16.84, and on Stock dataset increases from 4.084 to 5.214. The mIoU on Stock dataset also drops from 0.569 to 0.538. This confirms that explicit depth information is crucial for the model to reason about spatial layout and occlusion, rather than just relying on 2D visual patterns. However, even without depth, our model still outperforms most baselines (e.g., CSENet on OPA FID 17.51, Stock FID 8.925) due to the strong MLLM backbone and high-quality training data.

**Unified training pipeline** We further explore the possibility of training one unified model for both tasks. We randomly select half of the content recommendation instruction data, togetger with the full object placement instruction data, forming the unified training data. We also enable multiple image inputs by randomly selecting half of the multi-round object placement instruction data and turning them into single round QA by inserting all the objects simultaneously. The trained model even improves in object placement ability, reaching FID score 3.764 ($-0.32$) and meanIoU 0.579 ($+0.01$) on the Large Corpus of Stock Images Val Set. The performance of the unified model on content recommendation is a little bit lower than our specific model as in Tab. 1, but still much higher than the baselines. We also provide the ablation studies on the different training strategies of the unified model on the Stock Val Set in Tab. 4, which show similar trends to our specific model.

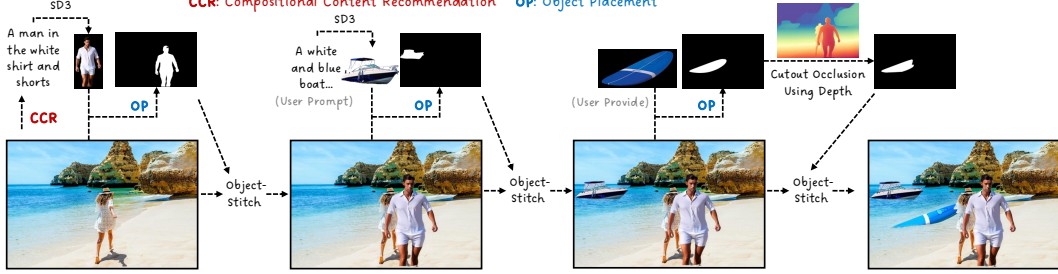

Figure 7: Illustration of a sequential image composition workflow. More details in Sec. 4.5.

Table 6: Analysis of Foreground Recommendation Diversity.

| Scene Type | Entropy (Ours) ↑ | Entropy (GT) | KL Divergence ↓ |
|---|---|---|---|
| Living Room | 4.12 | 4.25 | 0.18 |
| Street | 3.89 | 4.01 | 0.22 |
| Kitchen | 3.95 | 4.10 | 0.15 |
| **Average** | **3.99** | **4.12** | **0.18** |

## 4.5 SEAMLESS INTEGRATION WITH SYNTHESIS MODELS

We demonstrate how our Compositional Content Recommendation (CCR) and Object Placement (OP) models integrate seamlessly with off-the-shelf synthesis tools in a sequential image composition workflow (Fig. 7). Starting with a bg image, CCR suggests a compatible fg concept, such as "A man..." which is then converted into an image using SD3 (Esser et al., 2024). Our OP model predicts the composition bounding box relative to the bg, and an off-the-shelf segmentation (Qi et al., 2022a) generates an accurate object mask at the predicted location. The fg, bg, and mask are combined using the pretrained ObjectStitch (Song et al., 2022) synthesis model to produce the final composite image. Users can either rely on CCR to generate fg suggestions or manually input their own concepts via text or image, offering flexibility in the workflow, as shown in the $2^{nd}$ and $3^{rd}$ stages of Fig. 7. Also, since the OP model predicts the depth of the inserted object, we compute occlusion masks to accurately position the object behind existing elements in the bg, enabling seamless integration, as shown in the rightmost of Fig. 7. Details of the occlusion mask computation are in Appendix F.

## 4.6 ANALYSIS OF RECOMMENDATION DIVERSITY

To address concerns about potential spurious correlations (e.g., overfitting to frequent object-background pairs), we analyze the diversity of our model's foreground recommendations. We classify 1,000 test background images into common scene types (Living Room, Street, Kitchen) and compare the distribution of predicted objects against the ground truth. As shown in Tab. 6, our model achieves an entropy score of 3.99, comparable to the ground truth's 4.12, indicating high diversity. The low KL divergence (0.18) suggests our model captures the natural semantic distribution well without collapsing to a single mode.

## 5 CONCLUSION

We proposed an innovative framework for compositional content recommendation and object placement, leveraging MLLMs to achieve SOTA performance in both tasks. Our approach integrates a scalable data generation pipeline, targeted data augmentation, and independent projectors to extract and encode diverse inputs effectively. Trained on carefully constructed instruction datasets, our models demonstrated superior performance on benchmarks like OPA and a large stock corpus, as well as high user preference in studies.

## ETHICS STATEMENT

In this paper, we propose a large-scale instruction tuning dataset for image composition finetuning, and trained MLLM-based models that can perform foreground/background recommendation and recommend precise and reasonable composition information. Outputs of our trained models may occasionally contain inappropriate content inherited from the base model, and such outputs do not reflect the authors' views. We strictly adhere to the ICLR ethical research standards and applicable laws. To the best of our knowledge, this work complies with the General Ethical Principles.

## REPRODUCIBILITY STATEMENT

We follow the ICLR reproducibility standards and ensure the reproducibility of our work. The code of experiments on the OPA dataset and the self-created in-the-wild dataset used for evaluation is provided in an anonymous GitHub link mentioned in the abstract: `https://anonymous.4open.science/r/MGCR`. The detailed experimental settings, including hyperparameters and implementation steps, are documented in the paper and the Appendix.

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

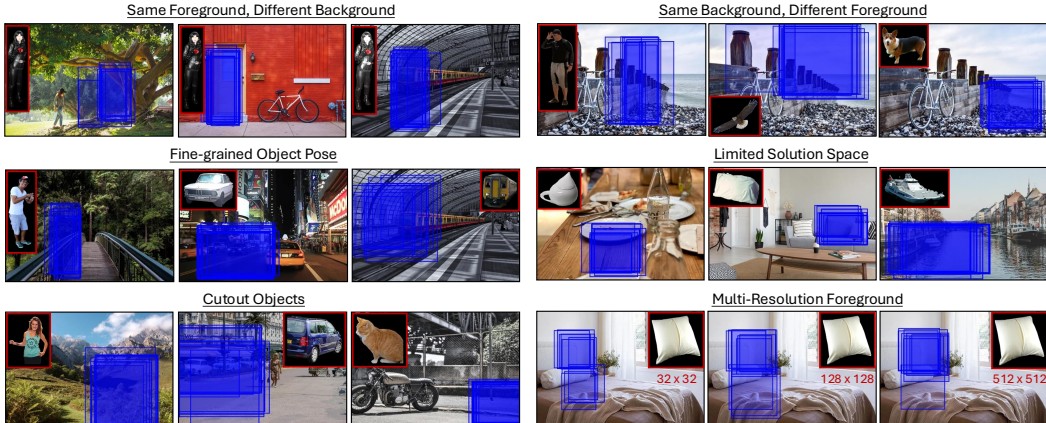

Figure 8: Qualitative comparisons of compositional content recommendation between state-of-the-arts MLLMs and ours.

Figure 9: Additional qualitative results of object placement on an in-the-wild test set across various scenarios.

## APPENDIX

## A QUALITATIVE COMPARISONS IN CONTENT RECOMMENDATION

We provide the qualitative comparisons in content recommendation tasks in Fig. 8. Our model shows obvious superiority over the compared baselines consistently.

## B SENSITIVITY TEST

We show the qualitative results of our object placement model across diverse testing scenarios in Fig. 9. For the left upmost example, when the given the same foreground object woman, our model can put her in reasonable locations with reasonable scale under different background scenarios.

For the right upmost example, when given the same background of seashore, our model can handle different types of input well. For the left middle example, our model can reflect the input foreground objects' poses to put them in appropriate places, which demonstrates its spatial detail understanding and reasoning ability. For the right middle example, when the placement solutions are limited, our model can put objects in the exact right spots. The "Limited Solution Space" means cases when the meaningful or compatible solutions are limited to certain place, e.g., the table area withput plates, the sofa and the river surface. For the left bottom example, given incomplete object inputs, our model is able to respond to the foreground cutout place to place the objects in places harmonious with the background boundaries. For the right bottom example, we show that our model can handle same input object with vastly different resolutions and output consistent bounding box suggestions. This is essential since MLLMs tend to overfit low level characteristics of image inputs in our experiments, and our data augmentation technique prevents such over-fitting problem and force the model to concentrate on the foreground background matching in semantics level.

We can draw the conclusion from Fig. 9 that our object placement model can offer reasonable and accurate placement suggestions consistenly, which demonstrates its potential in real-world applications and products.

---

**Algorithm 1** Random Mask Subset Selection

---

1: **Input:**
  - $M$: set of mask files $\{m_1, \ldots, m_n\}$ (n objects)
  - $p_{\text{proximity}}$: proximity selection probability
  - $r_{\text{initial}}$: initial selection ratio (default=0.4)

2: **Initialization:**
  - If $|M| \leq 2$: **return** M
  - $m_{\text{rand}} \leftarrow \text{RandomSelect}(M)$

3: **Mask Sorting:**
4: **if** $\text{rand}(0, 1) < p_{\text{proximity}}$ **then**
5:    **for** each $m_i$ in $M$ **do**
6:       $c_i \leftarrow \text{ComputeCenter}(m_i)$
7:       $d_i \leftarrow \|c_{\text{rand}} - c_i\|_2$
8:    **end for**
9:    $\pi \leftarrow \text{ArgsortAscending}(\{d_1, \ldots, d_n\})$
10: **else**
11:    $s_{\text{rand}} \leftarrow \text{ComputeSize}(m_{\text{rand}})$
12:    **for** each $m_i$ in $M$ **do**
13:       $s_i \leftarrow \text{ComputeSize}(m_i)$
14:       $\text{diff}_i \leftarrow |s_{\text{rand}} - s_i|$
15:    **end for**
16:    $\pi \leftarrow \text{ArgsortAscending}(\{\text{diff}_1, \ldots, \text{diff}_n\})$
17: **end if**
18: **Mask Grouping:**
19: $r \leftarrow r_{\text{initial}}$
20: **while** $|I| \leq 1$ **do**
21:    $k \leftarrow \max(1, \text{round}(r \times |M|))$
22:    $I \leftarrow \text{UniqueSort}(\text{first } k \text{ masks})$
23:    $r \leftarrow \min(r + 0.2, 1.0)$
24: **end while**
25: **return** I

---

## C    DETAILED DESCRIPTION OF OPA (LIU ET AL., 2021) DATASET

The OPA (Liu et al., 2021) dataset contains 62,074 training images (21,376 positive and 40,698 negative) and 11,396 testing images (3,588 positive and 7,808 negative). All images are sampled and selected from the COCO (Lin et al., 2014) dataset, and each composite image is composed of one background, one object, and one composition bounding box. Positive samples feature foreground and background compositions that are deemed compatible, while negative samples contain mismatched compositions.

## D    RANDOM SUBSET SELECTION ALGORITHM IN INPAINTING MASK MERGING

For each image, suppose there are $n$ object entities segmented out, resulting in $n$ entity segmentation masks in total: $\{m_1, \cdots, m_n\}$. if $n \leq 2$, there is no need to select the subset so we just adopt all the masks $M$. If $n > 2$, we first randomly select a mask denoted as $m_{\text{rand}}$, and then select the rest of the masks via location-based or size-based criteria. We define a proximity selection probability $p_{\text{proximity}}$ (default value 0.7).

With probability $p_{\text{proximity}}$, we obey location-based criteria for mask selection. Specifically, for each mask $m_i \in M$, we compute the geometric center of the mask area (the white pixel area) and calculate the distance $d_i$ w.r.t. the geometric center of $m_{\text{rand}}$. The mask selection priority list $\pi$ is then determined by the sorted list of $\{d_i\}$ with ascending order.

With probability $1 - p_{\text{proximity}}$, we follow the size-based criteria for mask selection. Specifically, for each mask $m_i \in M$, we compute the size (number of pixels) of the mask area (the white pixel area) and calculate the difference w.r.t. the size of the mask $\text{diff}_i$. The mask selection priority list $\pi$ is then determined by the sorted list of $\{\text{diff}_i\}$ with ascending order.

Having the mask selection priority list $\pi$ in hand, we can now do mask subset selection. Our aim is to select more than one mask (the randomly selected mask $m_{\text{rand}}$ must be selected already), and by default we select the top $r_{\text{initial}} = 40\%$ of the mask selection priority list. If this ratio does not give more than one selection, we add the ratio by $20\%$ in a loop. The detailed algorithm is shown in Alg. 1.

## E  DETAILED IMPLEMENTATION OF DATA AUGMENTATION AND SOME VISUALIZATIONS

For both background and foreground images, we adopt the data augmentation types: color jittering, brightness transform, and downsampling.

- For color jittering, we use the `transforms.ColorJitter` function from package `torchvision` and set the hyperparameters contrast$= 0.25$, saturation$= 0.25$ and hue$= 0.1$. We apply random color jittering for both background and foreground images.

- To make the brightness adjustment augmentation imitating uneven light projection from different directions and angles, we design the `brightness_transform` function by creating a linear gradient mask and scaling its intensity. Let the input image be $I(x, y)$, defined in the spatial domain with values normalized to $[0, 1]$, where $H$ and $W$ represent the height and width of the image, respectively. A linear gradient mask is generated using spatial coordinates $u(x) = \frac{x}{W-1}$ and $v(y) = \frac{y}{H-1}$ for each pixel $(x, y)$, combined as $g(x, y) = u(x)\cos(\theta) + v(y)\sin(\theta)$, where $\theta$ is a randomly chosen angle in the range $[0, 2\pi]$. This gradient is normalized to the range $[0, 1]$ as $g'(x, y) = \frac{g(x,y)-\min(g)}{\max(g)-\min(g)}$. To adjust the brightness, the normalized gradient is scaled to create a mask $M(x, y) = 0.7 + g'(x, y) \cdot 0.6$, which scales the brightness range to $[0.7, 1.3]$. The adjusted image is then computed as $I'(x, y) = \min\big(I(x, y) \cdot M(x, y), 1\big)$ to ensure all values remain within the valid range $[0, 1]$. This method effectively enhances or reduces brightness across the image based on the generated gradient. Therefore the augmented image can be written as

$$I'(x, y) = \min\left( I(x, y) \cdot \left( 0.7 + \frac{g(x, y) - \min(g)}{\max(g) - \min(g)} \cdot 0.6 \right), 1 \right),$$

  where

$$g(x, y) = u(x)\cos(\theta) + v(y)\sin(\theta),$$
$$u(x) = \frac{x}{W-1}, \quad v(y) = \frac{y}{H-1}.$$

  We apply the random brightness adjustment to both the background and foreground images.

- For the foreground images in the training data we use the highest resolution versions we can get, and for background images we use the version of data with short-edge resized to 1024. The resolutions for both the background and foreground images have been large enough for the testing phase so we only need to do downsampling for resolution adjustment data augmentation. Specifically, we adopt a continuous resolution value downsampling approach via resizing the long edge of each image to a random value between a pre-defined value (32 for foreground, 200 for background) and the minimum of the long edge dimension of the image and 332 (the CLIP (Radford et al., 2021) encoder resizing resolution). For background images we apply downsampling with probability 0.2 for each sample and for foreground images we apply downsampling with probability 0.5 for each sample.

For around $78\%$ of the background images, we randomly inpaint multiple background regions to introduce high-frequency artifacts throughout the image before background data augmentation. We

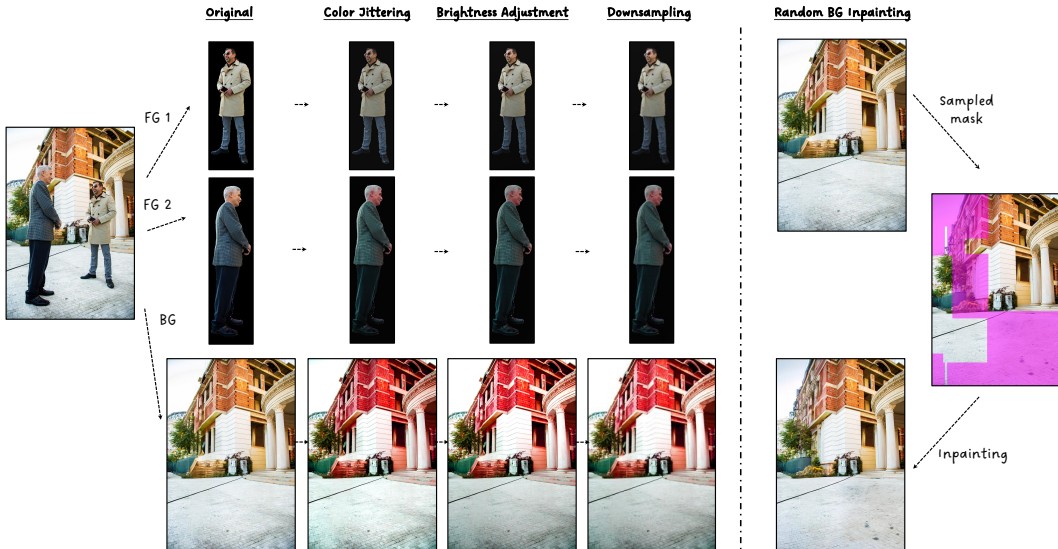

(a) Two men talking in front of a building. The image is disassembled into one background and two foreground images.

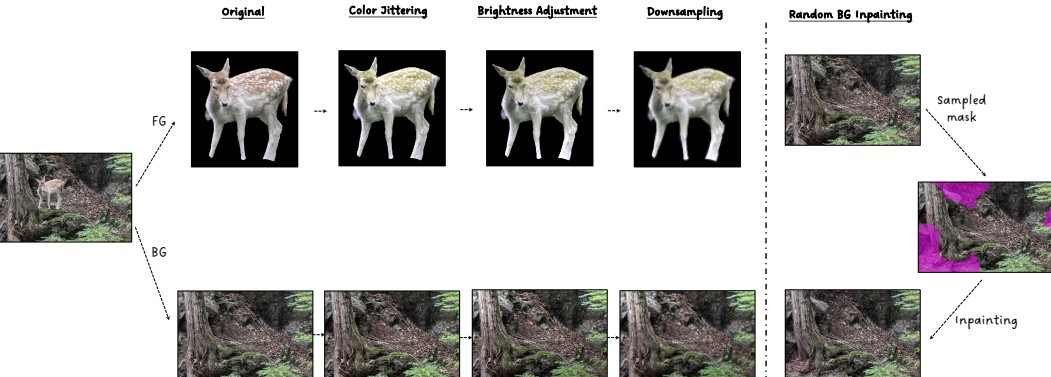

(b) One deer standing beside a tree. The image is disassembled into one background and one foreground image.

Figure 10: Two examples of training data augmentation. For each training sample, we sequentially apply random color jittering, random brightness adjustment and random downsampling (with pre-defined probability) for both the foreground images and background images. For some of the background images we conduct random inpainting (randomly sampled mask area in pink) before data augmentation. Because we resize to the same resolution for visualization, the effect of downsampling turns into blurring. This is actually the case for MLLM input since the CLIP vision encoder pad and resize all the input images to the $336 \times 336$ resolution. Best viewed in color.

sample some free-form masks similar to the ones used in Co-modulated GAN (Zhao et al., 2021) and the sampling mask size ratio is between $0.05$ and $0.9$.

We provide two data augmentation examples in the training data for visualization in Fig. 10. The aim of data augmentation is to break the low-level consistency of foreground images and background images in order to force the model to concentrate on image semantics in composition learning.

## F OCCLUSION MASK COMPUTATION

Our framework can generate object average depth prediction, which enables occlusion mask computation for inserting the object behind existing objects in the background image. To handle occlusions effectively, we compute the occlusion-adjusted object mask, denoted as $M_{O,\text{occluded}}$, based on

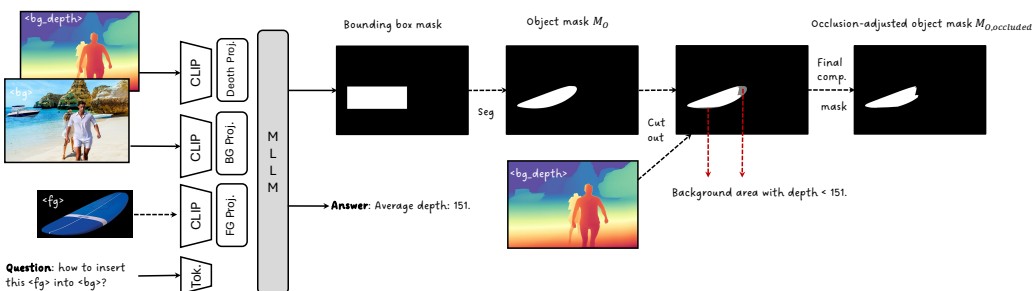

Figure 11: Illustration of the detailed occlusion mask computation process of the example in Fig. 7. More details are explained in Sec. F. Best viewed in color.

the predicted average depth value $d$ of the object and the depth map of the background $D_B$. Given the initial binary object mask $M_O$, where $M_O(x, y) = 1$ if the pixel $(x, y)$ belongs to the object and 0 otherwise, the occlusion condition is determined as $d > D_B(x, y)$, implying that the object should be behind the background at those pixels (farther to the camera). To update the mask, for each pixel $(x, y)$, we check if both $M_O(x, y) = 1$ (indicating the pixel belongs to the object) and $d > D_B(x, y)$. If both conditions are met, the corresponding pixel is wiped out from the object mask. Mathematically, the occlusion-adjusted foreground object mask is computed as:

$$M_{O,\text{occluded}}(x, y) = M_O(x, y) \cdot \mathbb{1}[d \leq D_B(x, y)],$$

where $\mathbb{1}[d \leq D_B(x, y)]$ is an indicator function which equals 1 if $d \leq D_B(x, y)$ and 0 otherwise. This formulation ensures that the object mask $M_{O,\text{occluded}}$ retains only the regions of the object where it is not occluded by the background. Fig. 11 illustrates the occlusion mask computation process of the example in Fig. 7.

## G  TRAINING DATA VISUALIZATION

We visualize four examples of the training bundles in Fig. 12. Each training bundle is composed of one background with related information and one or more foregrounds with related information. For background, the depth map is generated to RGB using Depth Anything-v2 (Yang et al., 2024b) form to be compatible with LLaVA (Liu et al., 2024c) model input image format. The background caption is generated using InternVL-1.5 (Chen et al., 2024c) by feeding the background image and ask question *"Provide a detailed description of the scene in the image."*. For foreground images, the captions are generated using VIP-LLaVA (Cai et al., 2024) via circling out the object using the entity segmentation mask (This local caption generation process is not $100\%$ accurate especially when encountering small objects, e.g., the carrot object in the 4th example in Fig. 12 is mislabeled as "A knife with a black handle." We leave room for improvement in foreground object captioning for future work). The location is computed from entity segmentation masks in the format of $(x, y, w, h)$ where the coordinate values are normalized to range $[0, 1000]$ w.r.t. the padded-to-square image. The average depth value is computed by calculating the average pixel depth value of the object area on the depth map of the original image. The context texts are generated via dividing the location information of each object into five types: top, bottom, left, right, middle, and dividing the depth information of each object into three types: near the camera, in the middle and distant from the camera. Corresponding controllable prompts are generated for instruction tuning based on the location and depth type the object sample is classified as. The instructions for both background and foreground are generated using 16 synonym prompts for background description and composition querying respectively.

## H  ADDITIONAL EXPERIMENTAL IMPLEMENTATION DETAILS

We develop our code base mainly based on the official GitHub repositories of LLaVA[1], GracoNet[2] and one unofficial implementation of TopNet (top). For the content recommendation task, we initial-

---

[1] https://github.com/haotian-liu/LLaVA
[2] https://github.com/bcmi/GracoNet-Object-Placement

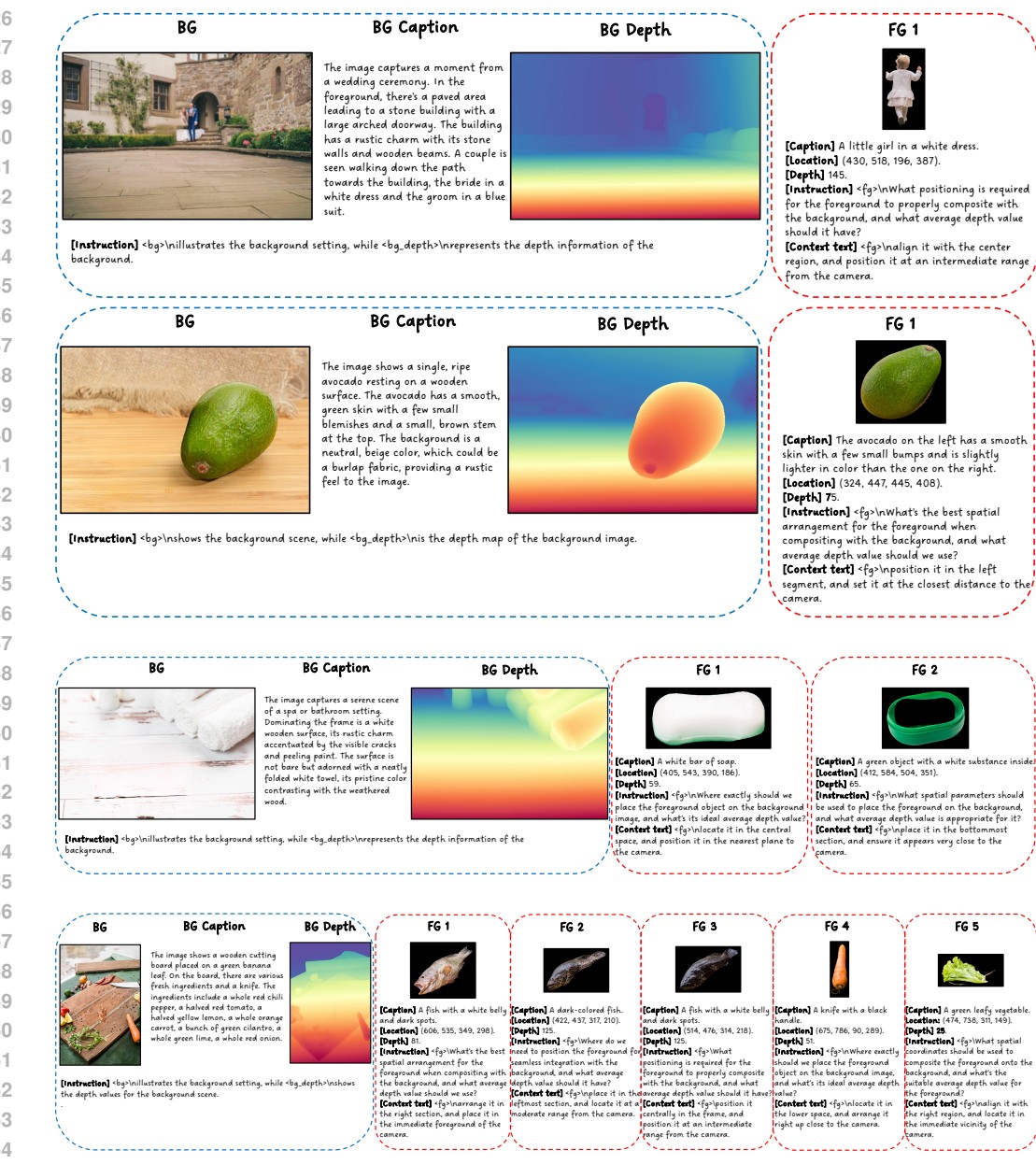

Figure 12: Four examples of training data bundles. Each training bundle is composed of background instruction, background image, background caption, background depth map, foreground image(s), average depth value(s) of foreground image(s), foreground bounding box(es), foreground caption(s), foreground instruction(s), and foreground context text(s). Best viewed in color.

ize our model from LLaVA-v1.6-13B and train for 1 epoch; for the object placement task, we initialize our model from LLaVA-v1.5-13B and train for 3 epochs. Following LLaVA, we adopt CLIP version clip-vit-large-patch14-336 for the vision encoder and a 2-layer MLPs with GeLU (Hendrycks & Gimpel, 2016) activation as projector architecture. We adopt AdamW (Loshchilov, 2017) optimizer for optimization with weight decay value 0. The learning rate is set as $2 \times 10^{-5}$ and we adopt the cosine annealing strategy with warmup ratio 0.03. The batch size is 8 per device. The training time is 54 hours for the content recommendation model and 165 hours for the object placement model. For inference, we set temperature $T = 0.2$ for all the experiments. All experiments are done

on NVIDIA A100-SXM4-80GB GPUs and Intel(R) Xeon(R) Platinum 8275CL CPU @ 3.00GHz CPUs with 96 logical processors.

For the Claude 3.7 Sonnet MLLM score on the Composition1K evaluation dataset, we use prompt *"Evaluate this image and give it a score from 0 to 100, indicating the degree of reasonability and plausibility of the image. Please focus on the reasonability of the placement of the foreground object(s) on the background. Explain your reasoning and provide the score at the end."*.

## I    COMPOSITION1K TEST SET

We include the anonymous Dropbox download link of Composition1K Test Set in `https://anonymous.4open.science/r/MGCR`. The dataset is composed of 195 scenes with 1149 image background foreground pairs in total.

## J    DATASHEET FOR DATASETS

We provide a detailed datasheet for our datasets used in the paper to discuss the potential biases and limitations.

**Motivation**

- **For what purpose was the dataset created?** The dataset was created to train MLLMs for generative composition recommendation, specifically for object placement and content recommendation tasks.
- **Who created the dataset?** The dataset was created by the authors of this paper using a scalable data generation pipeline.

**Composition**

- **What do the instances that comprise the dataset represent?** Each instance represents a background image, a foreground object (image or caption), and the corresponding placement information (bounding box and depth).
- **How many instances are there in total?** The training dataset contains 709K background images and over 1.3M background-foreground pairs.
- **Does the dataset contain all possible instances or is it a sample?** It is a sample of possible compositions, filtered from a larger corpus of stock images.

**Collection Process**

- **How was the data associated with each instance acquired?** The data was acquired from a large corpus of internal stock images.
- **If the dataset is a sample from a larger set, what was the sampling strategy?** We used MLLM-based filtering to exclude images with blank/uniform backgrounds or repetitive objects.
- **Who was involved in the data collection process?** The authors designed the automated pipeline; no external crowdworkers were directly involved in the *collection* (though crowdworkers annotated the original stock images).

**Biases and Limitations**

- **Demographic Bias:** The dataset relies on stock photography, which may contain inherent biases in representation (e.g., Western-centric scenes, stereotypical gender roles). Our model may learn and reproduce these biases (e.g., associating "nurse" with "female").
- **Scene Bias:** The filtering process explicitly removes crowded or distant scenes to focus on salient object placement. This creates a bias against these scene types, leading to lower performance on them (as discussed in Failure Cases).

- **Content Bias:** The dataset is filtered for "safe" content using MLLM tag checking, but some subtle biases or inappropriate associations might remain.

## K    COUNTERFACTUAL ROBUSTNESS ANALYSIS

To further quantify the robustness of our model, we conduct a counterfactual analysis by intentionally introducing noise to the input modalities. We test on a subset of 500 samples from the Stock Val Set.

- **Wrong Caption:** We replace the correct foreground caption with a random object caption. The placement mIoU drops by only $2.5\%$, suggesting the model relies heavily on visual features (RGB + Depth) and can often ignore semantic mismatches if the visual affordance is strong.
- **Noisy Depth:** We add Gaussian noise to the input depth map. The mIoU remains relatively stable ($-1.2\%$), but the occlusion accuracy (judged by humans) drops significantly ($-15\%$), confirming that depth is critical for *z-ordering* even if x-y placement is robust.
- **Wrong Background Caption:** Replacing the background caption has minimal impact ($-0.8\%$ mIoU), indicating the model extracts scene context primarily from the visual encoder.

## L    FAILURE MODE ANALYSIS

We categorize the primary failure modes of our system based on a manual analysis of 500 samples:

- **Depth Prediction Errors ($\sim$15%):** Inaccurate depth estimation can lead to incorrect occlusion masks (e.g., objects floating in front of foreground obstacles or cutting through the floor). This is the most significant source of visual artifacts.
- **Captioning Errors ($\sim$10%):** The MLLM occasionally misidentifies background objects or context, leading to semantically mismatched recommendations. However, the visual features often compensate for this, preserving reasonable placement.
- **Segmentation Artifacts:** Ideally, off-the-shelf segmentation should be perfect, but fuzzy boundaries can cause halo effects during compositing.

## M    PROMPT DETAILS FOR GPT-4O ON OPA TASK

We provide the detailed prompt design for using GPT-4o with In-Context Learning (ICL) on the OPA task. We select 5-shot examples from our training data. The prompt structure includes 5 examples followed by the query for the new image.

The raw code snippet for the 5-shot examples is as follows:

```
examples = [
    {
        "images": [
            "0000499eb8c07bfd55248e2e572b9f79/"
            "bg_perturb.jpg",
            "0000499eb8c07bfd55248e2e572b9f79/"
            "bg_depth.jpg",
            "0000499eb8c07bfd55248e2e572b9f79/"
            "fg_0_normalized_raw.png"
        ],
        "prompt": "represents the background, and\n"
                  "illustrates the depth data for the "
                  "background.\n"
                  "What positioning is required for the "
                  "foreground to properly composite with "
```

```
1188                        "the background?",
1189                "answer": "[213, 523, 426, 618]"
1190        },
1191        {
1192            "images": [
1193                "00003960e727d2cc9f9518ec064ff70b/"
1194                "bg_perturb.jpg",
1195                "00003960e727d2cc9f9518ec064ff70b/"
1196                "bg_depth.jpg",
1197                "00003960e727d2cc9f9518ec064ff70b/"
1198                "fg_0_normalized_raw.png"
1199            ],
1200            "prompt": "illustrates the background setting, "
1201                        "while\nrepresents the depth information "
1202                        "of the background.\n"
1203                        "Where exactly should we place the "
1204                        "foreground object on the background image?",
1205                "answer": "[662, 416, 74, 160]"
1206        },
1207        {
1208            "images": [
1209                "000069d8d4f54d994ead6b0af817dc77/"
1210                "bg_perturb.jpg",
1211                "000069d8d4f54d994ead6b0af817dc77/"
1212                "bg_depth.jpg",
1213                "000069d8d4f54d994ead6b0af817dc77/"
1214                "fg_0_normalized_raw.png"
1215            ],
1216            "prompt": "represents the background, and\n"
1217                        "is the depth map of the background image.\n"
1218                        "What are the coordinates for placing the "
1219                        "foreground on the background image?",
1220                "answer": "[462, 544, 263, 382]"
1221        },
1222        {
1223            "images": [
1224                "00003c9a0a8123136f767569d80c316a/"
1225                "bg_perturb.jpg",
1226                "00003c9a0a8123136f767569d80c316a/"
1227                "bg_depth.jpg",
1228                "00003c9a0a8123136f767569d80c316a/"
1229                "fg_0_normalized_raw.png"
1230            ],
1231            "prompt": "represents the background, and\n"
1232                        "illustrates the depth data for the "
1233                        "background.\n"
1234                        "What is the ideal placement of the "
1235                        "foreground object for compositing "
1236                        "with the background?",
1237                "answer": "[432, 308, 107, 412]"
1238        },
1239        {
1240            "images": [
1241                "000232f101eb3687bb55bb0d48de92d0/"
                "bg_perturb.jpg",
                "000232f101eb3687bb55bb0d48de92d0/"
                "bg_depth.jpg",
                "000232f101eb3687bb55bb0d48de92d0/"
                "fg_0_normalized_raw.png"
```

```
        ],
        "prompt": "illustrates the background setting, "
                  "while\nrepresents the depth information "
                  "of the background.\n"
                  "What's the best spatial arrangement "
                  "for the foreground when compositing "
                  "with the background?",
        "answer": "[297, 516, 282, 214]"
    }
]
```

The prompt is randomly selected from the following list for each example (16 variants with same meaning but different wording):

```
questions = [
    "What should be the spatial location of the foreground to be "
    "composited with the background? Please only provide the "
    "coordinates in the format of [x,y,w,h] with no other "
    "information, where x and y are the center coordinates and "
    "w and h are the width and height with respect to the "
    "background image extended to square. Please only give the "
    "answer in the format of '[x,y,w,h].'.",

    "Where should the foreground be positioned when combining it "
    "with the background? Please only provide the coordinates "
    "in the format of [x,y,w,h] with no other information, "
    "where x and y are the center coordinates and w and h are "
    "the width and height with respect to the background image "
    "extended to square. Please only give the answer in the "
    "format of '[x,y,w,h].'.",

    "What is the ideal placement of the foreground object for "
    "compositing with the background? Please only provide the "
    "coordinates in the format of [x,y,w,h] with no other "
    "information, where x and y are the center coordinates and "
    "w and h are the width and height with respect to the "
    "background image extended to square. Please only give the "
    "answer in the format of '[x,y,w,h].'.",

    "In which position should the foreground be located to blend "
    "with the background? Please only provide the coordinates "
    "in the format of [x,y,w,h] with no other information, "
    "where x and y are the center coordinates and w and h are "
    "the width and height with respect to the background image "
    "extended to square. Please only give the answer in the "
    "format of '[x,y,w,h].'.",

    "What are the coordinates for placing the foreground on the "
    "background image? Please only provide the coordinates in "
    "the format of [x,y,w,h] with no other information, where "
    "x and y are the center coordinates and w and h are the "
    "width and height with respect to the background image "
    "extended to square. Please only give the answer in the "
    "format of '[x,y,w,h].'.",

    "How should we spatially arrange the foreground relative to "
    "the background? Please only provide the coordinates in the "
    "format of [x,y,w,h] with no other information, where x and "
    "y are the center coordinates and w and h are the width and "
```

```
"height with respect to the background image extended to "
"square. Please only give the answer in the format of "
"'[x,y,w,h].'.",

"What is the optimal location for the foreground when merging "
"it with the background? Please only provide the coordinates "
"in the format of [x,y,w,h] with no other information, "
"where x and y are the center coordinates and w and h are "
"the width and height with respect to the background image "
"extended to square. Please only give the answer in the "
"format of '[x,y,w,h].'.",

"Where do we need to position the foreground for seamless "
"integration with the background? Please only provide the "
"coordinates in the format of [x,y,w,h] with no other "
"information, where x and y are the center coordinates and "
"w and h are the width and height with respect to the "
"background image extended to square. Please only give the "
"answer in the format of '[x,y,w,h].'.",

"What spatial coordinates should be used to composite the "
"foreground onto the background? Please only provide the "
"coordinates in the format of [x,y,w,h] with no other "
"information, where x and y are the center coordinates and "
"w and h are the width and height with respect to the "
"background image extended to square. Please only give the "
"answer in the format of '[x,y,w,h].'.",

"How should we determine the foreground's position when "
"combining it with the background? Please only provide the "
"coordinates in the format of [x,y,w,h] with no other "
"information, where x and y are the center coordinates and "
"w and h are the width and height with respect to the "
"background image extended to square. Please only give the "
"answer in the format of '[x,y,w,h].'.",

"What's the best spatial arrangement for the foreground "
"when compositing with the background? Please only provide "
"the coordinates in the format of [x,y,w,h] with no other "
"information, where x and y are the center coordinates and "
"w and h are the width and height with respect to the "
"background image extended to square. Please only give the "
"answer in the format of '[x,y,w,h].'.",

"Where exactly should we place the foreground object on the "
"background image? Please only provide the coordinates in "
"the format of [x,y,w,h] with no other information, where "
"x and y are the center coordinates and w and h are the "
"width and height with respect to the background image "
"extended to square. Please only give the answer in the "
"format of '[x,y,w,h].'.",

"What positioning is required for the foreground to properly "
"composite with the background? Please only provide the "
"coordinates in the format of [x,y,w,h] with no other "
"information, where x and y are the center coordinates and "
"w and h are the width and height with respect to the "
"background image extended to square. Please only give the "
"answer in the format of '[x,y,w,h].'.",
```

```
    "How can we specify the foreground's location for optimal "
    "compositing with the background? Please only provide the "
    "coordinates in the format of [x,y,w,h] with no other "
    "information, where x and y are the center coordinates and "
    "w and h are the width and height with respect to the "
    "background image extended to square. Please only give the "
    "answer in the format of '[x,y,w,h].'.",

    "What spatial parameters should be used to place the "
    "foreground on the background? Please only provide the "
    "coordinates in the format of [x,y,w,h] with no other "
    "information, where x and y are the center coordinates and "
    "w and h are the width and height with respect to the "
    "background image extended to square. Please only give the "
    "answer in the format of '[x,y,w,h].'.",

    "Which coordinates would best situate the foreground when "
    "overlaying it on the background? Please only provide the "
    "coordinates in the format of [x,y,w,h] with no other "
    "information, where x and y are the center coordinates and "
    "w and h are the width and height with respect to the "
    "background image extended to square. Please only give the "
    "answer in the format of '[x,y,w,h].'."
]
```

## N  RESULTS ON CARTOON IMAGES

We evaluate our composition recommendation model on three cartoon-style sample pairs. As shown in Fig. 13, our model can generate reasonable composition recommendations for the cartoon-style sample pairs.

## O  COMPARISON WITH NANO BANANA AND FLUX

We address the question of whether specialized OPA tasks are necessary given the rise of text-guided editing models like Nano Banana or Flux Context.

**1. Task Difference:**  Our model takes raw image inputs (bg + fg object) directly. In contrast, models like Nano Banana require a text description ("place a small dog on the ground"). Using them for our task would require an additional VLM to first caption the foreground and background, introducing an extra error source and latency.

**2. Out-painting Artifacts:**  Even with perfect captions, current text-guided editors often halluci-nate or expand the background borders to fit the object, rather than strictly adhering to the original background canvas. This distortion is unacceptable for precise graphic design workflows. (See Fig. 14 for examples).

**3. Composition Quality:**  We tested Nano Banana Pro on our Composition1K dataset. While impressive for generation, its spatial logic for specific object insertion often fails to respect scene affordances (e.g., placing objects floating or with incorrect scale relative to the depth). Fig. 15 illustrates these limitations.

## P  FAILURE CASES

In Fig. 16, we show some failure cases of our object placement model on the most challenging benchmark — our self-created Composition1K evaluation set. Our model fails in object placement

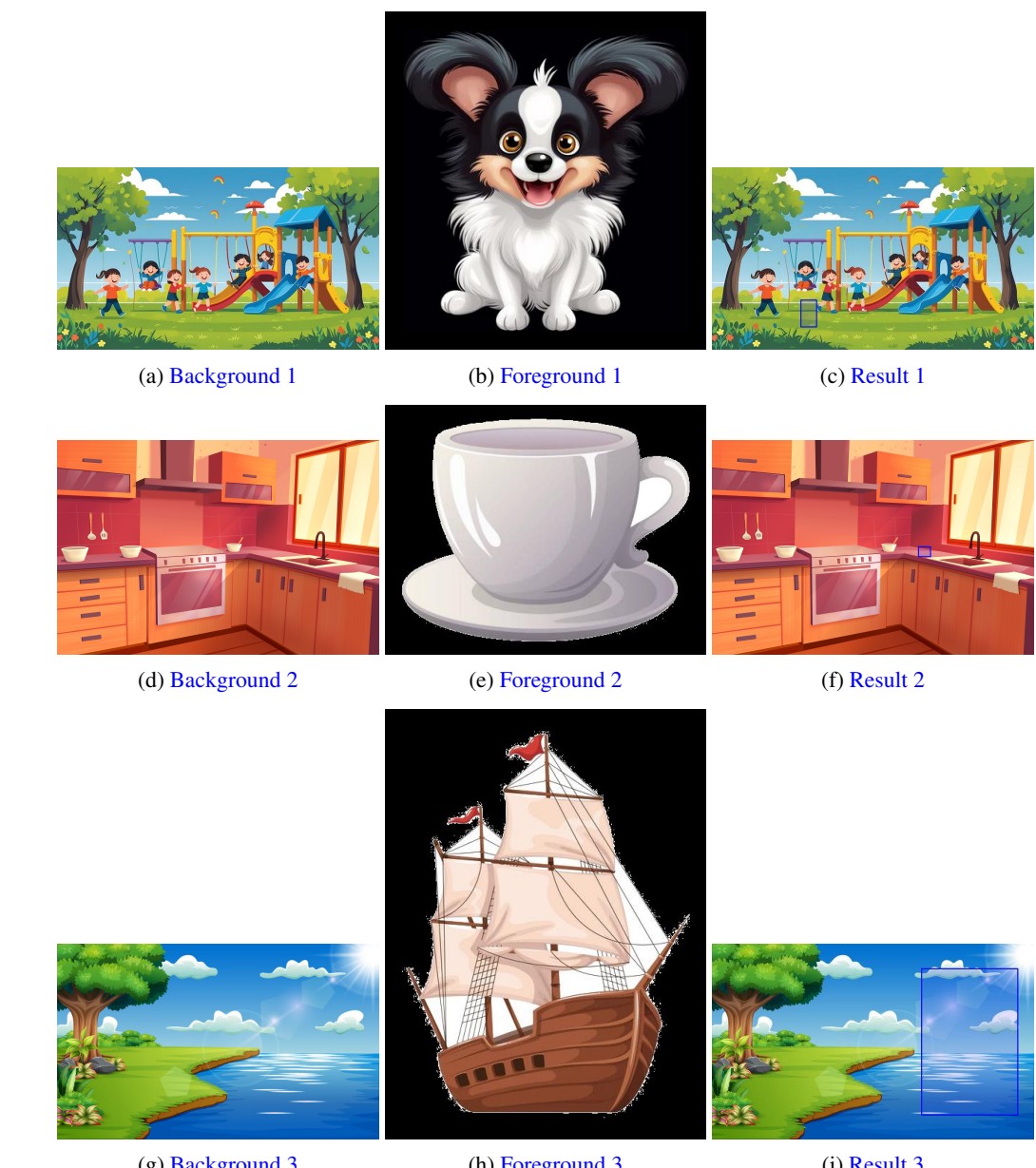

Figure 13: Examples of composition recommendations for cartoon-style sample pairs.

when the background image contain a lots of people or the background image is very distant from the camera. The suggested bounding boxes tend to be large and some of them are in unreasonable places.

This phenomenon might arises from the lack of such kind of backgrounds in our collected stock training data. Most of our stock images are near the camera. Also, images with lots of people, especially from a distant view, might have been filtered out in the MLLM-based filtering stage as the InternVL model will view a large number of small-size people as repetitive objects. To resolve such corner cases, we plan to curate more comprehensive training data bundles with more distant images. We leave this for the future work.

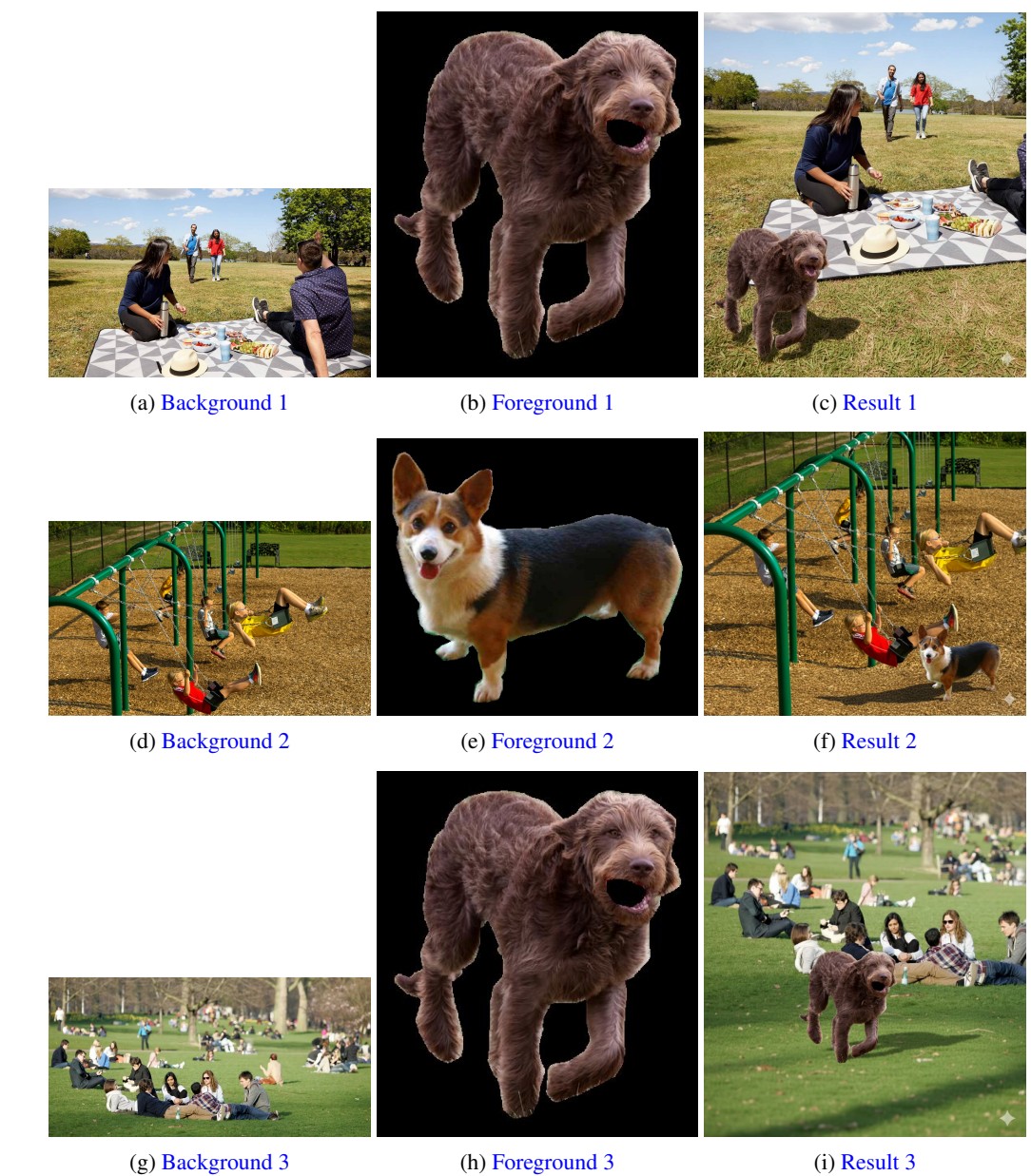

(a) Background 1     (b) Foreground 1     (c) Result 1

(d) Background 2     (e) Foreground 2     (f) Result 2

(g) Background 3     (h) Foreground 3     (i) Result 3

Figure 14: Examples of out-painting artifacts and border distortion from Nano Banana Pro when used for object placement. Each row shows the input background, input foreground, and the resulting composition.

Table 7: Per-sample inference time comparison on the Composition1K evaluation dataset.

| Methods | Average Inference Time (ms) |
|---|---|
| LLaVA-1.5-13B (Liu et al., 2024a) | 1778 |
| GPT-4o with ICL (OpenAI, 2024) | 5392 |
| Ours | 1849 |

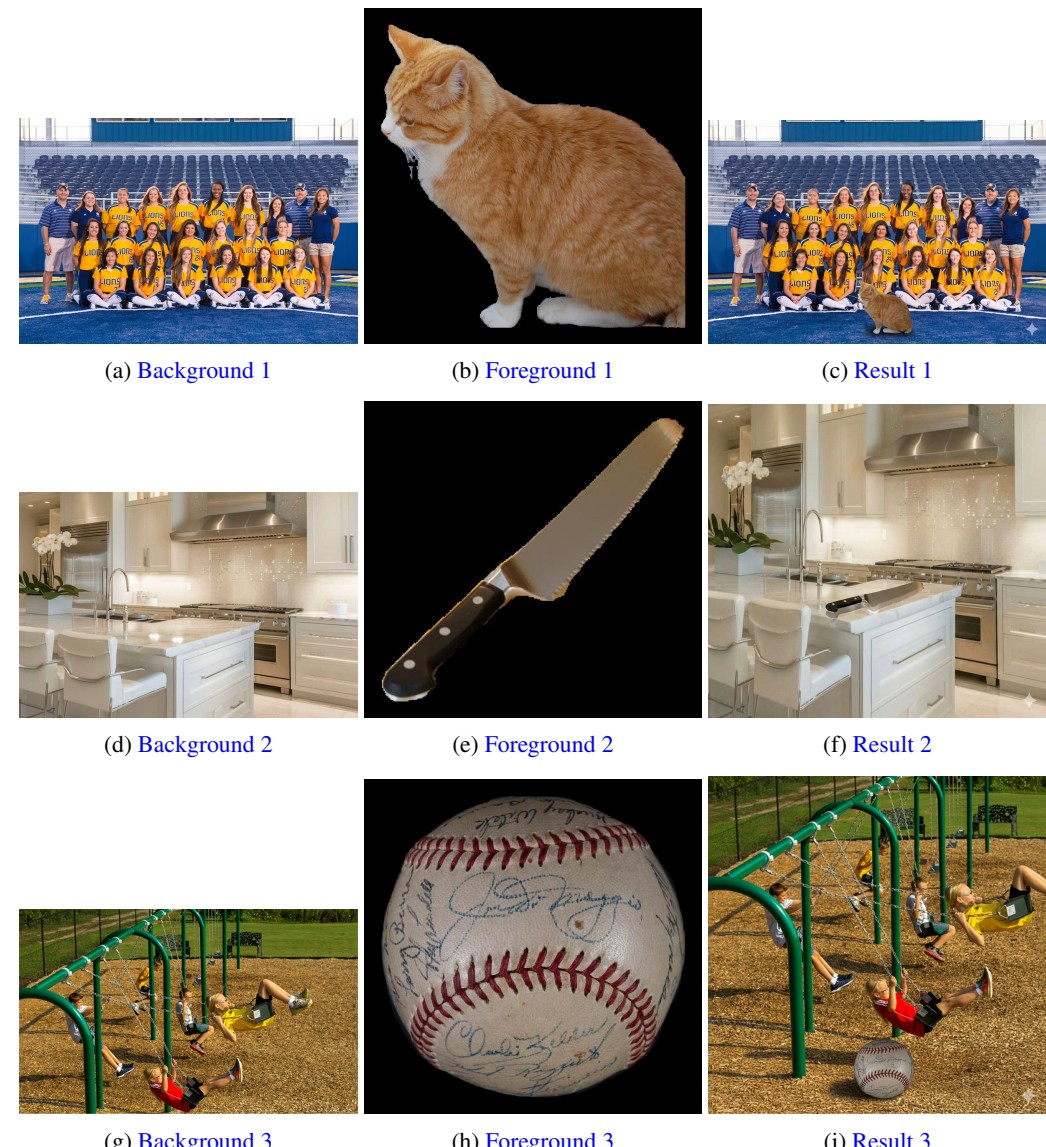

(a) Background 1  (b) Foreground 1  (c) Result 1

(d) Background 2  (e) Foreground 2  (f) Result 2

(g) Background 3  (h) Foreground 3  (i) Result 3

Figure 15: Examples of suboptimal composition (floating objects (the second row), scale mismatch (usually the object is too large, the first and the third row)) from text-guided editors. Each row shows the input background, input foreground, and the resulting composition.

## Q  INFERENCE SPEED

We test the average evaluation sample inference speed of three MLLM approaches compared in the main paper on our self-created Composition1K evaluation dataset. We compare our object placement model with LLaVA-1.5-13B and ChatGPT-4o using OPENAI API with in context learning. As illustrated in Tab. 7, our model spends 1.849s for inference per sample, which demonstrates considerable efficiency and practical feasibility.

## R  LLM USAGE

The use of LLMs is a general-purpose assist tool to aid or polish writing. We utilized GPT-5 to refine certain aspects of the writing in the Introduction and Related Works sections.

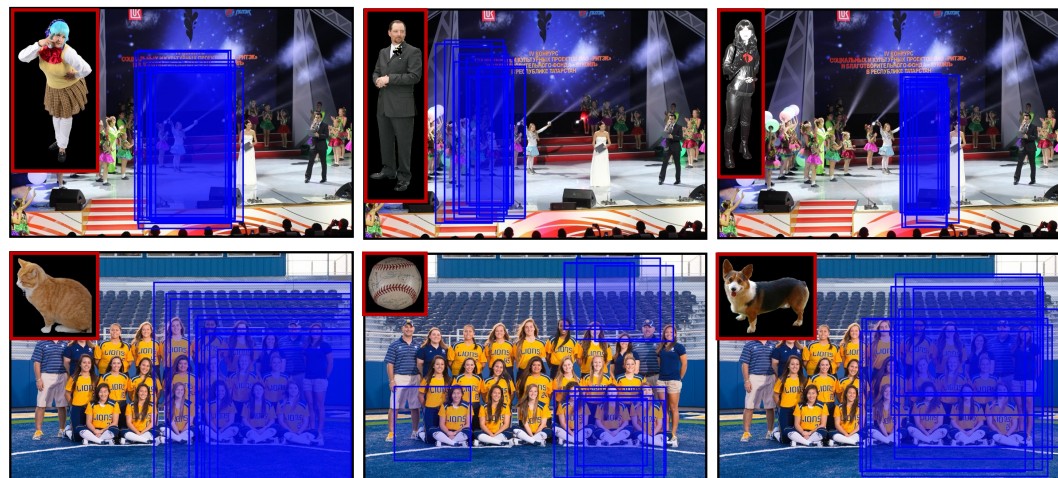

(a) Failure case examples of two backgrounds with many people. The first row shows a background of ceremony where there are quite a few people with relatively small sizes. Given input foregrounds of different people, our model tends to produce unreasonably large bounding boxes, depsite having comparatively reasonable locations. The second row shows a background of Women's Ball Team Family Portrait with many girls. Give inputs of cat, dog and a baseball, the suggested bounding boxes tend to be large in scale and also incompatible in locations.

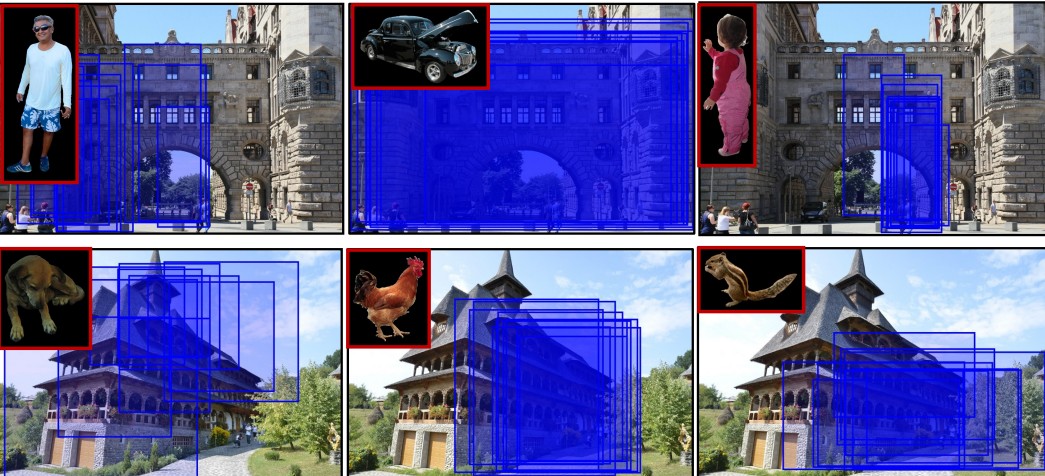

(b) Failure case examples of two backgrounds which are distant from the camera. The first row shows a background of a western ancient building which is compartively far away. Given input foregrounds of people and a car, our model tends to produce unreasonably large bounding boxes, depsite having comparatively reasonable locations. The second row shows a background of a east-Asian temple which is also far away from the camera. Give input foregrounds of middle to small size animals, the suggested bounding boxes tend to be very large in scale and also sometimes incompatible in locations.

Figure 16: Failure cases of our object placement model on the Composition1K evaluation dataset. (10 independent runs for each sample)

