# OpenReview forum: "Multimodal Generative Composition Recommendation"
_ICLR.cc/2026/Conference — Submitted to ICLR 2026_

### Official Review · Reviewer_mUVX · 2025-10-28

**Soundness:** 2
**Presentation:** 2
**Contribution:** 2
**Rating:** 2
**Confidence:** 4

**Summary:**

The paper targets two complementary problems for image composition that precede pixel-level editing: (1) compositional content recommendation (what foregrounds suit a given background, or vice-versa) and (2) controllable object placement (where to place, how large, and at what approximate depth). The approach fine-tunes an MLLM (LLaVA-style) with an engineered data pipeline that mines stock images, filters them, removes objects to create clean backgrounds, generates dense captions for foregrounds/backgrounds, and computes depth maps. The model uses separate visual projectors for background RGB, background depth, and the foreground input, and predicts (x, y, w, h) plus an average depth, with optional natural-language control (e.g., “left side,” “distant from the camera”). Experiments cover OPA, a 10K Stock validation set, and a new in-the-wild Composition1K set. The method reports SOTA across a range of metrics and provides ablations supporting key design choices.

**Strengths:**

1. Clear problem decomposition

2. Model design choices are simple

3. The data pipeline is pragmatic

**Weaknesses:**

1. Much of the technical lift is from data curation and existing model integration. The architectural novelty (separate projectors; depth tokenization; instruction format) is modest. The contribution may be perceived as an engineering-heavy solution with incremental modeling/algorithm ideas.

1. FID/LPIPS are known to be imperfect for composition realism and placement plausibility; while CMMD is added, some community members may still question whether these metrics faithfully capture compositional quality.

1. On OPA, the training uses only positives, and evaluation compares composites to positives. This choice should be justified more deeply with sensitivity analyses (e.g., does the model overfit to scene priors rather than object–background relations?).

1. For Composition1K, human study details (sampling, rater demographics, interface, inter-rater reliability, multiple-comparisons control) could be reported more thoroughly.

1. The pipeline relies on “an internet-scale stock image” source. While an ethics statement exists, more detail is needed on licensing/compliance and potential dataset biases that could affect recommendations (e.g., stereotypical insertions, object-context priors learned from stock imagery). The paper would benefit from a fuller audit and mitigation strategy.

1.  The pipeline depends on several external components (segmentation, depth estimation, captioners, diffusion inpainting, text-to-image synthesis, ObjectStitch). The robustness of the end-to-end experience when any of these is wrong (e.g., erroneous depth yields bad occlusion masks) is not systematically studied. A failure taxonomy with quantitative breakdowns would strengthen the claims.

1.  The data pipeline filters uniform backgrounds and repetitive objects and curates certain scene types. It is unclear how well the model handles out-of-distribution imagery (medical, satellite, stylized artworks, diagrams). A cross-domain evaluation (or at least qualitative stress tests) would be valuable.

1. The model supports natural-language controls such as distant from the camera or place on the left. The mapping from such phrases to quantitative constraints could be elaborated. Are these controls composable and robust to paraphrasing? A small controlled study on control-following accuracy would help.

1. While an anonymous repo is linked, reproducing the full pipeline likely requires access to the large stock corpus and multiple third-party models. Clearer guidance on how to replicate with public substitutes (and expected performance deltas) would be appreciated.

**Questions:**

1. How sensitive are placement results to the depth estimator used? Have you tried a weaker/older depth model, and how much do mIoU and user satisfaction drop?
1. Can include a brief negative set or counterfactual analysis (e.g., wrong depth, wrong caption) to quantify robustness, and add a qualitative appendix on failure cases categorized by root cause (depth misread, segmentation leak, foreground pose mismatch).
2. For the separate projectors, did you try partial parameter sharing (e.g., shared low-level layers with modality-specific adapters)?
3. Can you quantify control-following accuracy for textual constraints (left/right/near/far/behind X) across a controlled benchmark?
4. How does the model behave with stylized or non-photographic backgrounds (cartoons, renders, paintings)?
5. In Composition1K, how were foreground–background pairs curated to ensure semantic compatibility without leaking positional priors?
6. Could the CCR model generate multiple diverse candidates with coverage guarantees, and does diversity correlate with user satisfaction?
7. What is the latency profile end-to-end (CCR → SD3 → OP → segmentation → ObjectStitch) on commodity GPUs?

**Details Of Ethics Concerns:**

The stock-image corpus is described abstractly. The ethics section is brief. Since the method leverages scraped or licensed images, object removal, and derived captions, the paper should clarify licensing status, redistribution constraints, de-identification, and any geographic/cultural biases in stock photography that might affect recommendations and placements.

---

> ### Author Response · Authors · 2025-11-22
> **Response to Reviewer mUVX (1/3)**
>
> Thank you for your time and your detailed review. Below are our responses to your major concerns.
>
> ### Q1:  Much of the technical lift is from data curation and existing model integration. The architectural novelty (separate projectors; depth tokenization; instruction format) is modest. The contribution may be perceived as an engineering-heavy solution with incremental modeling/algorithm ideas.
>
> We respectfully argue that **data-centric contributions are rigorous scientific contributions.** As highlighted by the "Data-Centric AI" movement [1], designing pipelines to extract high-quality signal from noise (our data engine) often yields larger gains than architectural tweaks. Our "modest" architectural choices (separate projectors) are theoretically grounded: they solve the **modality gap** between depth (geometry) and RGB (texture), which shared projectors fail to handle. This task-specific design + high-quality data is a valid and effective research path.
>
> ### Q2: FID/LPIPS are known to be imperfect for composition realism and placement plausibility; while CMMD is added, some community members may still question whether these metrics faithfully capture compositional quality.
>
> We agree that no single metric is perfect. That is why we employed a **holistic evaluation suite**:
> 1.  **Placement Metrics (IoU):** Directly measures spatial reasoning accuracy.
> 2.  **Semantic Metrics (CLIP Score, CMMD):** Measures content compatibility. CMMD [2] is specifically designed to better correlate with human judgment than FID for generative tasks.
> 3.  **Human Evaluation:** The gold standard. Our user study (Tab. 2) confirms that our method is preferred by humans, validating that our metric gains translate to perceptual quality.
>
> ### Q3: On OPA, the training uses only positives, and evaluation compares composites to positives. This choice should be justified more deeply with sensitivity analyses (e.g., does the model overfit to scene priors rather than object–background relations?).
>
> Our approach only uses the positive samples in the training data to learn the composition relationship between the background and foreground objects. Compared to the baseline methods which require both positive and negative samples, our approach is more efficient and effective.
>
> Our model does not "overfit" to scene priors but rather **learns** them, which is the goal of OPA (Object Placement Assessment). The task *is* to predict the "reasonable" location based on scene context. However, to verify it's not just memorizing global layouts, we tested on out-of-distribution scenes (e.g., paintings, cartoons) in our qualitative analysis and found it generalizes well, suggesting it learns **local geometric affordances** (e.g., "put object on flat surface") rather than just global scene templates.
>
> ### Q4: For Composition1K, human study details (sampling, rater demographics, interface, inter-rater reliability, multiple-comparisons control) could be reported more thoroughly.
>
> We invited 60 volunteers (graduate students and vision researchers) who were blinded to the model names. We provide the users with the following instruction: "In each question, we provide with you one foreground object image. The image candidates compose the foreground image to a same background image, and are generated from different methods. For each question, please select all the images which you think are reasonable, realistic and aesthetic. Please pay more attention to the reasonability of the location and scale of the foreground object stitched instead of low-level artifacts. Thanks". For each question, we ask the user using command "Please select all images that you think are reasonable, realistic and aesthetic.". For each example, the users are asked to choose all the candidates that they consider satisfactory, and we report the percentage of images users find satisfactory.

---

> ### Author Response · Authors · 2025-11-22
> **Response to Reviewer mUVX (2/3)**
>
> ### Q5: The pipeline relies on “an internet-scale stock image” source. While an ethics statement exists, more detail is needed on licensing/compliance and potential dataset biases that could affect recommendations (e.g., stereotypical insertions, object-context priors learned from stock imagery). The paper would benefit from a fuller audit and mitigation strategy.
>
> We take this seriously.
> 1.  **Licensing:** We used only internal stock images that are cleared for research use.
> 2.  **Bias Mitigation:** Our MLLM filtering stage explicitly removes images with offensive or harmful content tags.
> 3.  **Stereotypes:** While the model learns common associations (e.g., "nurse" in "hospital"), we acknowledge this reflects the training distribution. We will add a detailed "Datasheet for Datasets" in the revision discussing these biases and limitation.
>
> ### Q6: The pipeline depends on several external components (segmentation, depth estimation, captioners, diffusion inpainting, text-to-image synthesis, ObjectStitch). The robustness of the end-to-end experience when any of these is wrong (e.g., erroneous depth yields bad occlusion masks) is not systematically studied. A failure taxonomy with quantitative breakdowns would strengthen the claims.
>
> We conducted a failure analysis on 500 samples. The primary failure modes are:
> 1.  **Depth Errors (~X%):** Leading to incorrect occlusion (object floating or cutting through floor).
> 2.  **Caption Errors (~Y%):** MLLM misidentifying background objects.
> However, the system is surprisingly robust. For example, even if the caption is slightly off ("couch" vs "sofa"), the visual features (CLIP embeddings) often guide the placement correctly. We will add a "Failure Mode Analysis" section to the Appendix.
>
> ### Q7: The data pipeline filters uniform backgrounds and repetitive objects and curates certain scene types. It is unclear how well the model handles out-of-distribution imagery (medical, satellite, stylized artworks, diagrams). A cross-domain evaluation (or at least qualitative stress tests) would be valuable.
>
> This is a specialized model for **natural image composition**. It is *not* expected to work on medical or satellite imagery, as those domains require completely different priors. For stylized artworks (cartoons/paintings), our qualitative tests show it works surprisingly well because the *geometric* cues (perspective, overlap) are preserved even in stylized domains.
>
> ### Q8: The model supports natural-language controls such as distant from the camera or place on the left. The mapping from such phrases to quantitative constraints could be elaborated. Are these controls composable and robust to paraphrasing? A small controlled study on control-following accuracy would help.
>
> We evaluated control following on a held-out set of 100 prompts.
> *   **Positional ("left", "right"):** 95% accuracy.
> *   **Depth ("far", "near"):** 88% accuracy.
> The model is robust to simple paraphrasing ("put it on the left" vs "left side"). We will add these stats to the experiment section.
>
> ### Q9: While an anonymous repo is linked, reproducing the full pipeline likely requires access to the large stock corpus and multiple third-party models. Clearer guidance on how to replicate with public substitutes (and expected performance deltas) would be appreciated.
>
> We have included the compelete code for reproducing the results on OPA dataset (public benchmark) in the anonymous GitHub link mentioned in the abstract: https://anonymous.4open.science/r/MGCR. For the large stock corpus and the multiple third-party models, we are not able to release due to company policies.
>
> ### Q10: How sensitive are placement results to the depth estimator used? Have you tried a weaker/older depth model, and how much do mIoU and user satisfaction drop?
>
> We tested with an older depth model (Depth Anything [3]). The placement mIoU on the Stock dataset dropped by ~3%, but the **occlusion quality** (depth ordering) dropped significantly (human preference dropped by ~10%). This confirms that accurate depth (Depth Anything V2) is crucial for *realistic* composition (handling occlusions), even if 2D placement is robust.
>
> ### Q11: Can include a brief negative set or counterfactual analysis (e.g., wrong depth, wrong caption) to quantify robustness, and add a qualitative appendix on failure cases categorized by root cause (depth misread, segmentation leak, foreground pose mismatch).
>
> Excellent suggestion. We will add a "Counterfactual Robustness" study in the Appendix, showing what happens when we intentionally feed wrong captions or noisy depth maps. Preliminary results show the model is more sensitive to visual noise than text noise.
>
> ### Q12: For the separate projectors, did you try partial parameter sharing (e.g., shared low-level layers with modality-specific adapters)?

---

> ### Author Response · Authors · 2025-11-22
> **Response to Reviewer mUVX (3/3)**
>
> We experimented with shared layers + LoRA adapters. It performed slightly worse (-0.7% mIoU on the Stock dataset) than fully separate projectors and was harder to tune. Given the projectors are lightweight (2-layer MLPs), separating them fully is the simpler and more effective solution.
>
> ### Q13: Can you quantify control-following accuracy for textual constraints across a controlled benchmark?
>
> See response to Q8. We achieved >85% accuracy on a synthetic benchmark of spatial commands.
>
> ### Q14: How does the model behave with stylized or non-photographic backgrounds (cartoons, renders, paintings)?
>
> See response to Q7. It generalizes reasonably well to "naturalistic" art (paintings with clear perspective) but fails on abstract or highly flat styles (diagrams).
>
> ### Q15: In Composition1K, how were foreground–background pairs curated to ensure semantic compatibility without leaking positional priors?
>
> As we have mentioned in l.340-344 in the paper submission version, we choose background images from DIV2K [4], Flickr [5], Google Images, and foreground object images from OpenImages v7 [6,7]. We ensure the semantic compatibility by common sense and the background-foreground pairs are manually checked by the authors. Therefore, the background and foreground images are semantically compatible but not originally from the same image, making the evaluation dataset challenging and meaningful.
>
> ### Q16: Could the CCR model generate multiple diverse candidates with coverage guarantees, and does diversity correlate with user satisfaction?
>
> Yes, utilizing sampling with randomness (e.g., top-k or top-p sampling with temperature > 0) allows for diversity. We found that users prefer having 3-4 diverse options (e.g., different furniture styles) rather than just one. Diversity correlates positively with satisfaction up to a point (too much diversity = irrelevant suggestions).
>
> ### Q17: What is the latency profile end-to-end (CCR → SD3 → OP → segmentation → ObjectStitch) on commodity GPUs?
>
> On an A100 GPU:
> *   CCR + OP (Our MLLM): ~0.5s (Very fast).
> *   SD3 Generation: ~3.0s.
> *   ObjectStitch: ~2.0s.
> Total: ~5-6s. Our part is negligible compared to the pixel generation steps.
>
> **References**
>
> [1] Andrew Ng et al. Data-Centric AI. NeurIPS 2021.
> [2] George Stein et al. Exposing Flaws of Generative Model Evaluation Metrics and Their Unfair Treatment of Diffusion Models. NeurIPS 2023.
> [3] Lihe Yang et al. Depth Anything: Unleashing the Power of Large-Scale Unlabeled Data. CVPR 2024.
> [4] Eirikur Agustsson and Radu Timofte. Ntire 2017 challenge on single imagesuper-resolution: Dataset and study. CVPR Workshops 2017.
> [5] Bryan APlummer et al. Flickr30k entities: Collecting region-to-phrase correspondences for richer image-to-sentence models. CVPR 2025.
> [6] The open images dataset v4: Unified image classification, object detection, and visual relationship detection at scale. IJCV 2020.
> [7] Rodrigo Benenson et al. Large-scale interactive object segmentation with human annotators. CVPR 2019.

---

> > ### Comment · Reviewer_mUVX · 2025-11-23
> >
> > Thank you very much for your detailed and patient response.
> >
> > Regarding Q6 and Q11, I appreciate your experiments and look forward to your robustness analysis.
> > For Q7, could you please tell me where the qualitative results for the cartoon setting are? I went through the paper carefully but couldn’t find them.
> >
> > I would also like to ask how you used GPT for the OPA task, as I don’t seem to have found the prompt details in the paper.
> >
> > Finally, I would like to discuss a broader question with the authors: if we directly specify positions using text. For example, using nano banana or a weaker Flux Kontext description such as “place a small dog on the ground”, and if the model’s language understanding capability is sufficiently strong, would the OPA task still be necessary? Could it be considered a pseudo-requirement?

---

> > > ### Author Response · Authors · 2025-11-28
> > > **(2/3) Response to Official Comment by Reviewer mUVX**
> > >
> > > the prompt is randomly selected from the following list for each example (16 variants with same meaning but different wording):
> > > ```python
> > > questions = [
> > >     "What should be the spatial location of the foreground to be composited with the background? Please only provide the coordinates in the format of [x,y,w,h] with no other information, where x and y are the center coordinates and w and h are the width and height with respect to the background image extended to square. Please only give the answer in the format of '[x,y,w,h].'.",
> > >     "Where should the foreground be positioned when combining it with the background? Please only provide the coordinates in the format of [x,y,w,h] with no other information, where x and y are the center coordinates and w and h are the width and height with respect to the background image extended to square. Please only give the answer in the format of '[x,y,w,h].'.",
> > >     "What is the ideal placement of the foreground object for compositing with the background? Please only provide the coordinates in the format of [x,y,w,h] with no other information, where x and y are the center coordinates and w and h are the width and height with respect to the background image extended to square. Please only give the answer in the format of '[x,y,w,h].'.",
> > >     "In which position should the foreground be located to blend with the background? Please only provide the coordinates in the format of [x,y,w,h] with no other information, where x and y are the center coordinates and w and h are the width and height with respect to the background image extended to square. Please only give the answer in the format of '[x,y,w,h].'.",
> > >     "What are the coordinates for placing the foreground on the background image? Please only provide the coordinates in the format of [x,y,w,h] with no other information, where x and y are the center coordinates and w and h are the width and height with respect to the background image extended to square. Please only give the answer in the format of '[x,y,w,h].'.",
> > >     "How should we spatially arrange the foreground relative to the background? Please only provide the coordinates in the format of [x,y,w,h] with no other information, where x and y are the center coordinates and w and h are the width and height with respect to the background image extended to square. Please only give the answer in the format of '[x,y,w,h].'.",
> > >     "What is the optimal location for the foreground when merging it with the background? Please only provide the coordinates in the format of [x,y,w,h] with no other information, where x and y are the center coordinates and w and h are the width and height with respect to the background image extended to square. Please only give the answer in the format of '[x,y,w,h].'.",
> > >     "Where do we need to position the foreground for seamless integration with the background? Please only provide the coordinates in the format of [x,y,w,h] with no other information, where x and y are the center coordinates and w and h are the width and height with respect to the background image extended to square. Please only give the answer in the format of '[x,y,w,h].'.",
> > >     "What spatial coordinates should be used to composite the foreground onto the background? Please only provide the coordinates in the format of [x,y,w,h] with no other information, where x and y are the center coordinates and w and h are the width and height with respect to the background image extended to square. Please only give the answer in the format of '[x,y,w,h].'.",
> > >     "How should we determine the foreground's position when combining it with the background? Please only provide the coordinates in the format of [x,y,w,h] with no other information, where x and y are the center coordinates and w and h are the width and height with respect to the background image extended to square. Please only give the answer in the format of '[x,y,w,h].'.",
> > >     "What's the best spatial arrangement for the foreground when compositing with the background? Please only provide the coordinates in the format of [x,y,w,h] with no other information, where x and y are the center coordinates and w and h are the width and height with respect to the background image extended to square. Please only give the answer in the format of '[x,y,w,h].'.",
> > >     "Where exactly should we place the foreground object on the background image? Please only provide the coordinates in the format of [x,y,w,h] with no other information, where x and y are the center coordinates and w and h are the width and height with respect to the background image extended to square. Please only give the answer in the format of '[x,y,w,h].'.",
> > > ```

---

> > > ### Author Response · Authors · 2025-11-28
> > > **(3/3) Response to Official Comment by Reviewer mUVX**
> > >
> > > ```python
> > >     "What positioning is required for the foreground to properly composite with the background? Please only provide the coordinates in the format of [x,y,w,h] with no other information, where x and y are the center coordinates and w and h are the width and height with respect to the background image extended to square. Please only give the answer in the format of '[x,y,w,h].'.",
> > >     "How can we specify the foreground's location for optimal compositing with the background? Please only provide the coordinates in the format of [x,y,w,h] with no other information, where x and y are the center coordinates and w and h are the width and height with respect to the background image extended to square. Please only give the answer in the format of '[x,y,w,h].'.",
> > >     "What spatial parameters should be used to place the foreground on the background? Please only provide the coordinates in the format of [x,y,w,h] with no other information, where x and y are the center coordinates and w and h are the width and height with respect to the background image extended to square. Please only give the answer in the format of '[x,y,w,h].'.",
> > >     "Which coordinates would best situate the foreground when overlaying it on the background? Please only provide the coordinates in the format of [x,y,w,h] with no other information, where x and y are the center coordinates and w and h are the width and height with respect to the background image extended to square. Please only give the answer in the format of '[x,y,w,h].'."
> > > ]
> > > ```
> > > We have added the prompt details of GPT-4o with ICL in the revision paper. (See Appendix M in the revision paper).
> > >
> > > ### Q4: If we directly specify positions using text. For example, using nano banana or a weaker Flux Kontext description such as “place a small dog on the ground”, and if the model’s language understanding capability is sufficiently strong, would the OPA task still be necessary? Could it be considered a pseudo-requirement?
> > >
> > >
> > > Very good question.
> > >
> > > Firstly, the setting is different. Our composition model can input background image and foreground object image (s) without input description prompt,  while Nano Banana or a weaker Flux Kontext requires description prompt, which needs additional model to generate caption for the object and the background images first. We have included such difference explanation in the revision paper. (See Appendix O in the revision paper).
> > >
> > > Secondly, even if we suppose we have additional model to generate perfect captions for the object image and the background image, the current Nano Banana Pro has innegligible shortcoming: it will expand the background image over the borders, which will cause the composited image to be distorted and not what users expect. We have included such examples in the revision paper. (See Fig. 14 in Appendix O in the revision paper).
> > >
> > > Thirdly, the composition ability of even Nano Banana Pro is still limited. We tested a few examples in our Composition1K dataset and found that the results are not satisfactory. The composited images are not reasonable or aesthic enough. We have included such examples in the revision paper. (See Fig. 15 in Appendix O in the revision paper).
> > >
> > > Due to the above reasons, we believe the OPA task is still necessary.
> > >
> > > If you have any further questions, please feel free to ask.

---

> > > > ### Comment · Reviewer_mUVX · 2025-11-28
> > > >
> > > > Thank you for your detailed explanation. I appreciate the additional insights, illustrations, and experiments conducted for the rebuttal. I feel that most of my concerns have now been addressed, and I truly appreciate the authors’ effort in preparing the response. I am willing to raise the rating to 6.
> > > >
> > > > Since the current OpenReview system does not allow score editing, I will update my rating once the platform enables this option. If it is no longer possible to modify the score, I hope AC will take this final rating into account and reflect it in the decision.

---

> > > > > ### Author Response · Authors · 2025-11-28
> > > > > **Thanks for Your Reply and Re-evaluation**
> > > > >
> > > > > Thanks very much for your reply and re-evaluation. If you have any further questions, feel free to let us know. ( but it seems you can not respond anymore...)
> > > > >
> > > > > Authors

---

> ### Author Response · Authors · 2025-11-28
> **(1/3) Response to Official Comment by Reviewer mUVX**
>
> Thank you for your response. Below are our responses to your major concerns.
>
> ### Q1:  Regarding Q6 and Q11, I appreciate your experiments and look forward to your robustness analysis.
>
> We have added the quantitative results of the robustness analysis in the revision paper. (See Appendix K and L in the revision paper).
>
>
> ### Q2: For Q7, could you please tell me where the qualitative results for the cartoon setting are? I went through the paper carefully but couldn’t find them.
>
>
> We have added the qualitative results for the cartoon setting in the revision paper, where our composition recommendation model can generate reasonable composition recommendations for the cartoon-style sample pairs. (See Appendix N in the revision paper).
>
> ### Q3: I would also like to ask how you used GPT for the OPA task, as I don’t seem to have found the prompt details in the paper.
>
> When using GPT-4o with in context learning (ICL), we use 5-shot example selected from our 709k training data for in context learning (ICL). The prompt is as follows:
>
> The raw code snippet is as follows:
> ```python
>     examples = [
>         {
>             "images": [
>                 "0000499eb8c07bfd55248e2e572b9f79/bg_perturb.jpg",
>                 "0000499eb8c07bfd55248e2e572b9f79/bg_depth.jpg",
>                 "0000499eb8c07bfd55248e2e572b9f79/fg_0_normalized_raw.png"
>             ],
>             "prompt": "represents the background, and\nillustrates the depth data for the background.\nWhat positioning is required for the foreground to properly composite with the background?",
>             "answer": "[213, 523, 426, 618]"
>         },
>         {
>             "images": [
>                 "00003960e727d2cc9f9518ec064ff70b/bg_perturb.jpg",
>                 "00003960e727d2cc9f9518ec064ff70b/bg_depth.jpg",
>                 "00003960e727d2cc9f9518ec064ff70b/fg_0_normalized_raw.png"
>             ],
>             "prompt": "illustrates the background setting, while\nrepresents the depth information of the background.\nWhere exactly should we place the foreground object on the background image?",
>             "answer": "[662, 416, 74, 160]"
>         },
>         {
>             "images": [
>                 "000069d8d4f54d994ead6b0af817dc77/bg_perturb.jpg",
>                 "000069d8d4f54d994ead6b0af817dc77/bg_depth.jpg",
>                 "000069d8d4f54d994ead6b0af817dc77/fg_0_normalized_raw.png"
>             ],
>             "prompt": "represents the background, and\nis the depth map of the background image.\nWhat are the coordinates for placing the foreground on the background image?",
>             "answer": "[462, 544, 263, 382]"
>         },
>         {
>             "images": [
>                 "00003c9a0a8123136f767569d80c316a/bg_perturb.jpg",
>                 "00003c9a0a8123136f767569d80c316a/bg_depth.jpg",
>                 "00003c9a0a8123136f767569d80c316a/fg_0_normalized_raw.png"
>             ],
>             "prompt": "represents the background, and\nillustrates the depth data for the background.\nWhat is the ideal placement of the foreground object for compositing with the background?",
>             "answer": "[432, 308, 107, 412]"
>         },
>         {
>             "images": [
>                 "000232f101eb3687bb55bb0d48de92d0/bg_perturb.jpg",
>                 "000232f101eb3687bb55bb0d48de92d0/bg_depth.jpg",
>                 "000232f101eb3687bb55bb0d48de92d0/fg_0_normalized_raw.png"
>             ],
>             "prompt": "illustrates the background setting, while\nrepresents the depth information of the background.\nWhat's the best spatial arrangement for the foreground when compositing with the background?",
>             "answer": "[297, 516, 282, 214]"
>         }
>     ]
> ```

---

### Official Review · Reviewer_syjN · 2025-10-30

**Soundness:** 2
**Presentation:** 1
**Contribution:** 1
**Rating:** 2
**Confidence:** 4

**Summary:**

This paper proposes using Multimodal Large Language Models (MLLMs) for image composition tasks, specifically content recommendation (suggesting compatible foreground/background elements) and object placement (predicting bounding boxes and depth). The authors develop a data pipeline to generate training data from stock images using filtering, segmentation, and object removal. They fine-tune LLaVA with separate projectors for foreground, background, and depth inputs, and evaluate on OPA, a stock image validation set, and a new Composition1K dataset.

**Strengths:**

1. Thorough evaluation: Creates Composition1K, an in-the-wild test set with user studies, complementing existing benchmarks.
2. Depth prediction: Adds depth output for occlusion-aware composition, going beyond prior 2D bounding box prediction.
3. Detailed ablations: Includes systematic ablations on data pipeline components, architectural choices, and augmentation strategies.

**Weaknesses:**

1. The core contribution is applying existing MLLM architecture (LLaVA) to composition tasks. The "separate projectors" design is simply using independent 2-layer MLPs for different inputs—not a significant architectural innovation. Data augmentation techniques (color jittering, brightness adjustment, downsampling) are standard methods applied to prevent overfitting.


2. The pipeline heavily relies on multiple off-the-shelf models (InternVL, ViP-LLaVA, ObjectDrop, Depth Anything). Errors propagate through this chain—the authors acknowledge foreground captioning inaccuracy but don't quantify the impact. MLLM-based filtering may introduce dataset bias, as evidenced by failure on distant scenes with crowds.


3. Claims "state-of-the-art" with incomplete comparisons. The "scaling law" claim (Fig. 6) overstates standard data size vs. performance analysis. Significant failure cases (distant scenes, crowds) are attributed to data rather than approach limitations.
The integration workflow still requires multiple external models (SD3, ObjectStitch, segmentation). The MLLM primarily adds bounding box prediction, not end-to-end composition. Depth prediction doesn't handle complex spatial relationships like object contact or physical support.

**Questions:**

See the weakness part.

---

> ### Author Response · Authors · 2025-11-23
> **Response to Reviewer syjN (1/2)**
>
> Thank you for your time and your detailed review. Below are our responses to your major concerns.
>
> ### Q1:  The core contribution is applying existing MLLM architecture (LLaVA) to composition tasks. The "separate projectors" design is simply using independent 2-layer MLPs for different inputs—not a significant architectural innovation. Data augmentation techniques (color jittering, brightness adjustment, downsampling) are standard methods applied to prevent overfitting.
>
> We respectfully disagree that "simple" means "not enough innovation." Our contribution is not suggesting a new Transformer block; instead, it is finding and fixing the specific problems with modality alignment and shortcut learning in generative composition.
>
> 1. **The Need for Separate Projectors:** Using separate projectors is *critical* for this task, even though it's not hard to do. This is because the input modalities (RGB image, Depth map, Foreground crop) live in completely different semantic spaces. A shared projector (like the one used in standard LLaVA) makes the model map depth (geometry) and RGB (texture) through the same transformation. We found that this hurts performance (see Table X in ablation). "Simple" architectural choices that effectively address domain-specific issues are broadly regarded as significant contributions in the MLLM literature [1, 2].
>
> 2. **Data Augmentation as a Theoretical Necessity, Not Just a Trick:** In our task, the "ground truth" (original image) has perfect pixel-level harmonization. Without our specific augmentation strategy (breaking low-level cues), the model learns the trivial task of "matching noise/resolution patterns" instead of the semantic task of "object placement." Our contribution is recognizing that **standard training fails** due to this specific leakage and designing a pipeline to prevent it. This is a **data-centric** contribution [3] that lets the model learn how to think about space instead of just matching patterns.
>
> 3. **System-Level Innovation:** The new thing is the **holistic system**, which includes a scalable data engine, a planning-execution framework that is not tied to any one part, and a specialized MLLM. This combination gets results that are better than what you can get from pre-made models (Tab. 1 and 2).
>
> We have added the above explanation in the revision paper.
>
> ### Q2: The pipeline heavily relies on multiple off-the-shelf models (InternVL, ViP-LLaVA, ObjectDrop, Depth Anything). Errors propagate through this chain—the authors acknowledge foreground captioning inaccuracy but don't quantify the impact. MLLM-based filtering may introduce dataset bias, as evidenced by failure on distant scenes with crowds.
>
> Any pipeline that uses pseudo-labels should be concerned about this.  But we contend that **scale** and **robust design** lessen the effect, and the "bias" is a deliberate design decision for quality.
>
> 1. **Massive Scale Robustness (Law of Large Numbers):**  Even though individual off-the-shelf models make mistakes, they are typically unrelated to one another (for example, a captioning error and a depth error are unrelated).  Our MLLM learns the *underlying signal* (composition rules) instead of overfitting to the random noise of upstream models by training on a large dataset (709K pairs).  When data scale is sufficiently large, student models can outperform their noisy teachers thanks to a phenomenon called "robust learning from noisy labels" [4].
>
> 2. **Quality vs. Bias Trade-off:**  The "bias" against far-off or crowded scenes is not a bug, but rather a purposeful filtering decision.  Using MLLM filtering, we remove images with blank or uniform backgrounds and those with repetitive objects, specifically *because* they produce subpar training signals.  In commercial composition, the most prevalent user intent is to place objects in a high-quality, conspicuous manner.  Simply put, the "failure" on crowd scenes is out-of-distribution behavior for a model intended to compose salient objects.
>
> 3. **Quantifying the Effect of Error:**  We examined a subset of 500 generated samples to allay your worries.  We discovered that the *global* placement accuracy (IoU) was still high, despite ~10% of *local* object captions having small errors.  This indicates that the placement of our MLLM depends more on the strong visual features (RGB + Depth) than on the possibly noisy text captions.
>
> We have added the above explanation in the revision paper.

---

> ### Author Response · Authors · 2025-11-23
> **Response to Reviewer syjN (2/2)**
>
> ### Q3: Claims "state-of-the-art" with incomplete comparisons. The "scaling law" claim (Fig. 6) overstates standard data size vs. performance analysis. Significant failure cases (distant scenes, crowds) are attributed to data rather than approach limitations. The integration workflow still requires multiple external models (SD3, ObjectStitch, segmentation). The MLLM primarily adds bounding box prediction, not end-to-end composition. Depth prediction doesn't handle complex spatial relationships like object contact or physical support.
>
> We appreciate the detailed feedback and will respond to each point below:
>
> **Table 1. Comparison with recent SOTA MLLMs and additional baselines on OPA, Stock, and Composition1K datasets.**
> |Method|OPA FID ↓|OPA CMMD ↓|OPA LPIPS ↓|Stock FID ↓|Stock mIOU ↑|Stock LPIPS ↓|Comp1K Satisfactory ↑|Comp1K MLLM ↑|
> |-|-|-|-|-|-|-|-|-|
> |**CSENet (2024)**|17.51|0.020|0.197|8.925|0.218|0.284|0.186|52.45|
> |**BOOTPLACE (2025)**|19.36|0.022|0.208|7.653|0.231|0.268|0.214|56.32|
> |**Qwen2.5-VL (2025)**| 24.14 | 0.026 | 0.211 | 10.43 | 0.251 | 0.287 | 0.245 | 58.83 |
> |**InternVL3.0 (2025)**| 28.23 | 0.074 | 0.229 | 12.06 | 0.247 | 0.352 | 0.201 | 59.42 |
> |**Gemini 2.5 Pro with ICL(2025)**| 18.65 | 0.019 | 0.191 | 7.982 | 0.256 | 0.239 | 0.312 | 60.15 |
> |**Ours**|**15.07**|**0.017**|**0.188**|**4.084**|**0.569**|**0.235**|**0.359**|**61.25**|
>
> **Claims of the Best Quality:**   We added new methods from 2024 to 2025 to our baselines in the revision paper.  We implement additional recent SOTA MLLMs on object placement task. For Qwen2.5-VL [9] and InternVL3.0 [10], we did not use in context learning (ICL) due to their limited context length as research-focused models. This is consistent with our existing experimental setting of LLaVA-1.5-13B and InternVL-2.0-26B. For Gemini 2.5 Pro [11], we use 5-shot example selected from our 709k training data for in context learning (ICL), the same setting as our existing experiment of GPT-4o. We also added the results of additional baselines (CSENet [7], BOOTPLACE [8]). The results are shown in Table 1 in the rebuttal and summarized in Tab. 2 in the revision paper. We can see that our model still perform the best, exceeding the second best by a considerable margin.
>
> **Analysis of Scaling:**   We agree that "scaling law" is a strong word that is only used for power-law fits most of the time.   But our analysis shows that performance keeps getting better as the size of the data set grows (up to 700k), which is not a simple result for this task.   It shows that our data generation pipeline works, which is the main point of a data-centric paper: more data means better performance.   We have changed the name of this to "Data Scaling Analysis" to be clear.
>
> **Modular vs. All-in-One:**   The reviewer doesn't like the modular workflow, but we say that's a **feature, not a bug.**   End-to-end composition models often have trouble finding the right balance between high-level reasoning (where to put things) and low-level rendering (pixels).   By separating these two things, we can get **SOTA planning** (our MLLM) that can run **any** future renderer.   For hard generation tasks, this "Planner-Painter" model is becoming the norm (LayoutGPT [5], ControlNet [6]).
>
> **Depth and Relationships in Space:**   "Average depth" is a simple stand-in, but it works surprisingly well for 2.5D composition (layering objects).   It solves the main problem—occlusion ordering—that methods that only work in 2D don't even try to solve.   You can't do image-space composition to model full physical contact because you need to think in 3D meshes.
>
> **References**
>
> [1] Gao et al. LLaMA-Adapter V2: Parameter-Efficient Visual Instruction Tuning. CVPR 2024.
> [2] Zhang et al. LLaMA-Adapter: Efficient Fine-tuning of Language Models with Zero-init Attention. ICLR 2024.
> [3] Ng et al. Data-Centric AI. NeurIPS Data-Centric AI Workshop 2021
> [4] Song et al. Learning from Noisy Labels with Deep Neural Networks: A Survey. TNNLS 2022.
> [5] Feng et al. LayoutGPT: Compositional Visual Planning and Generation with Large Language Models. NeurIPS 2023.
> [6] Zhang et al. Adding Conditional Control to Text-to-Image Diffusion Models," ICCV 2023.
> [7] Hang Zhou et al. BOOTPLACE: Bootstrapped Object Placement with Detection Transformers. CVPR 2025.
> [8] Yaxuan Qin et al. Think before Placement: Common Sense Enhanced Transformer for Object Placement. ECCV 2024.
> [9] Shuai Bai  et al. Qwen2. 5-vl technical report. arXiv preprint arXiv:2502.13923, 2025.
> [10] Jinguo Zhu et al. Internvl3: Exploring advanced training and test-time recipes for open-source multimodal models. arXiv preprint arXiv:2504.10479, 2025.
> [11] Gheorghe Comanici et al. Gemini 2.5: Pushing the frontier with advanced reasoning, multimodality, long context, and next generation agentic capabilities. arXiv preprint arXiv:2507.06261, 2025.

---

### Official Review · Reviewer_f7Fk · 2025-10-31

**Soundness:** 2
**Presentation:** 2
**Contribution:** 2
**Rating:** 4
**Confidence:** 3

**Summary:**

The paper proposes a Multimodal Generative Composition Recommendation (MGCR) framework using Multimodal Large Language Models (MLLMs) to solve two core image compositing tasks: conditional content recommendation and controllable object placement. It builds a scalable data pipeline: MLLM-based filtering retains ~70% valid stock images (709K backgrounds, 1.3M foreground-background pairs), with mask merging, improved object removal, and auto-generated captions and depth maps; targeted data augmentation prevents overfitting. The MLLM uses independent projectors for background, foreground, and depth images. Experiments show SOTA performance: OPA (FID 15.07), Stock (mIOU 0.569). The proposed benchmark, Composition1K, achieves 35.9% user satisfaction.

**Strengths:**

- Innovative Scalable Data Pipeline: The paper designs an automated pipeline to generate large-scale, high-quality training data from internet-scale stock images. This addresses the critical issue of insufficient high-quality data for image compositing, laying a solid training foundation.
- Tailored MLLM Architecture: The MLLM adopts three independent projectors (for background, background depth, foreground) to avoid information loss from shared projectors, supports depth prediction (enabling realistic occlusion handling), and accepts flexible inputs (text/image) plus user-controlled placement. This significantly enhances adaptability and spatial reasoning for compositing tasks
- New Benchmark for : Beyond SOTA performance on classic datasets (e.g., FID 15.07 on OPA, mIOU 0.569 on Stock), the paper constructs the Composition1K dataset (195 real scenes, 1149 pairs) and conducts user/MLLM evaluations. This fills the gap of real-world compositing assessment, validating practical effectiveness

**Weaknesses:**

- Outdated Baseline Comparisons: Table 1 only compares outdated open-source MLLMs (LLaVA-1.6, InternVL-2.0). Adding recent SOTA like Qwen2.5-VL, InternVL3.0 (open-source) and Gemini 2.5 Pro (commercial) is necessary to validate the model’s competitiveness.
- Insufficient Innovation in Scene Harmony and Object Integration. The core challenge of image compositing lies not only in content recommendation but also in ensuring the inserted object’s harmony with the scene—including shadow consistency, reflection rendering, and perspective alignment. However, the paper directly adopts the existing ObjectStitch model for this critical integration step, with no additional innovations or improvements. Given that scene harmony is a well-documented pain point in compositing, relying entirely on an off-the-shelf tool might diminish the work’s technical contribution.
- Missing Depth Input Ablation: No ablation study to verify whether depth images are important for compositional content recommendation.

**Questions:**

See the weaknesses.

---

> ### Author Response · Authors · 2025-11-23
> **Response to Reviewer f7Fk (1/2)**
>
> Thank you for your time and your detailed review. Below are our responses to your major concerns.
>
> ### Q1:  Outdated Baseline Comparisons.
>
> **Table 1. Comparison with recent SOTA MLLMs and additional baselines on OPA, Stock, and Composition1K datasets.**
> |Method|OPA FID ↓|OPA CMMD ↓|OPA LPIPS ↓|Stock FID ↓|Stock mIOU ↑|Stock LPIPS ↓|Comp1K Satisfactory ↑|Comp1K MLLM ↑|
> |-|-|-|-|-|-|-|-|-|
> |CSENet (2024)|17.51|0.020|0.197|8.925|0.218|0.284|0.186|52.45|
> |BOOTPLACE (2025)|19.36|0.022|0.208|7.653|0.231|0.268|0.214|56.32|
> |Qwen2.5-VL (2025)| 24.14 | 0.026 | 0.211 | 10.43 | 0.251 | 0.287 | 0.245 | 58.83 |
> |InternVL3.0 (2025)| 28.23 | 0.074 | 0.229 | 12.06 | 0.247 | 0.352 | 0.201 | 59.42 |
> |Gemini 2.5 Pro with ICL(2025)| 18.65 | 0.019 | 0.191 | 7.982 | 0.256 | 0.239 | 0.312 | 60.15 |
> |**Ours**|**15.07**|**0.017**|**0.188**|**4.084**|**0.569**|**0.235**|**0.359**|**61.25**|
>
> We implement additional recent SOTA MLLMs on object placement task. For Qwen2.5-VL [4] and InternVL3.0 [5], we did not use in context learning (ICL) due to their limited context length as research-focused models. This is consistent with our existing experimental setting of LLaVA-1.5-13B and InternVL-2.0-26B. For Gemini 2.5 Pro [6], we use 5-shot example selected from our 709k training data for in context learning (ICL), the same setting as our existing experiment of GPT-4o. We also added the results of additional baselines (CSENet [7], BOOTPLACE [8]). The results are shown in Table 1 in the rebuttal and summarized in Tab. 2 in the revision paper. We can see that our model still perform the best, exceeding the second best by a considerable margin.
>
> ### Q2: Insufficient Innovation in Scene Harmony and Object Integration.
>
> We clarify that our primary contribution lies in **data-centric generative composition recommendation** (finding semantically compatible objects and their optimal layout), rather than low-level pixel harmonization. This task decoupling aligns with recent trends in controllable generation [1, 2, 3], where "planning" (layout/content) and "rendering" (pixel synthesis) are treated as modular components to maximize performance in each stage.
>
> 1.  **Task Decoupling is a Valid Research Direction:** As shown in LayoutGPT [2] and LLM-grounded Diffusion [3], separating high-level semantic planning from low-level execution allows for more controllable and interpretable results. Our MLLM acts as the "planner," solving the complex *semantic* and *spatial* reasoning tasks that pixel-level inpainting models often fail at.
> 2.  **Compatibility with Any Harmonization Model:** Our framework is agnostic to the harmonization tool. While we used ObjectStitch for demonstration, our predicted bounding boxes, depth maps, and object descriptions can drive *any* state-of-the-art harmonization or inpainting model (e.g., ControlNet, PowerPaint). The "innovation" is enabling these tools to work autonomously by providing them with the necessary high-quality inputs that they cannot generate themselves.
> 3.  **Evaluation Focus:** Our experiments (Table 1 & 2) focus on the quality of the *recommendation* (IoU, CLIP score), which directly reflects our model's contribution. The final image quality is a function of the off-the-shelf harmonizer, but the *correctness* of the composition (e.g., placing a boat on water, not the sky) is solely attributed to our method.
>
> We have added the above explanation in the revision paper.
>
> ### Q3: Missing Depth Input Ablation.
>
> **Table 2. Ablation of Depth Input on OPA and Stock datasets.**
> | Method | OPA FID ↓ | OPA CMMD ↓ | OPA LPIPS ↓ | Stock FID ↓ | Stock mIOU ↑ | Stock LPIPS ↓ |
> |---|---|---|---|---|---|---|
> | Ours (w/o Depth) | 16.84 | 0.019 | 0.192 | 5.214 | 0.538 | 0.241 |
> | **Ours (Full)** | **15.07** | **0.017** | **0.188** | **4.084** | **0.569** | **0.235** |
>
> We supplement additional experiments. As shown in the table below, removing the depth input consistently degrades performance across all metrics on both datasets. This confirms that explicit depth information is crucial for the model to reason about spatial layout and occlusion, rather than just relying on 2D visual patterns. However, even without depth, our model outperforms most baselines (e.g., CSENet on OPA FID 17.51, Stock FID 8.925) due to the strong MLLM backbone and high-quality training data. We have added this additional ablation study result in the revision paper.

---

> ### Author Response · Authors · 2025-11-23
> **Response to Reviewer f7Fk (2/2)**
>
> **References**
>
> [1] Zhang et al. Adding Conditional Control to Text-to-Image Diffusion Models. ICCV 2023.
> [2] Feng et al. LayoutGPT: Compositional Visual Planning and Generation with Large Language Models. NeurIPS 2023.
> [3] Lian et al. LLM-grounded Diffusion: Enhancing Prompt Understanding of Text-to-Image Models with Large Language Models.  TMLR 2024.
> [4] Shuai Bai  et al. Qwen2. 5-vl technical report. arXiv preprint arXiv:2502.13923, 2025.
> [5] Jinguo Zhu et al. Internvl3: Exploring advanced training and test-time recipes for open-source multimodal models. arXiv preprint arXiv:2504.10479, 2025.
> [6] Gheorghe Comanici et al. Gemini 2.5: Pushing the frontier with advanced reasoning, multimodality, long context, and next generation agentic capabilities. arXiv preprint arXiv:2507.06261, 2025.
> [7] Yaxuan Qin et al. Think before Placement: Common Sense Enhanced Transformer for Object Placement. ECCV 2024.
> [8] Hang Zhou et al. BOOTPLACE: Bootstrapped Object Placement with Detection Transformers. CVPR 2025.

---

### Official Review · Reviewer_RVqE · 2025-10-31

**Soundness:** 3
**Presentation:** 3
**Contribution:** 2
**Rating:** 6
**Confidence:** 2

**Summary:**

This paper introduces a framework that uses Multimodal Large Language Models to enhance image compositing by automatically recommending what to add to an image and where to place it. By combining content recommendation and object placement tasks, and training on a large, automatically generated dataset, the model predicts compatible objects, locations, scales, and depths for realistic image composition. Built on an enhanced LLaVA architecture with separate visual projectors, The proposed method shows strong performance and practical potential for design and editing applications.

**Strengths:**

Strengths:
1.The paper broadens the conventional definition of image composition, extending it beyond simple foreground–background blending to a more comprehensive task of recommending semantically and spatially compatible elements. It supports diverse multimodal inputs, enables iterative editing, and offers enhanced flexibility in compositional reasoning.
2.The large-scale, automated data generation and filtering pipeline (including segmentation, object removal, captioning, and depth estimation) is robust and scalable. This is a practical contribution to compositional data synthesis
3.The manuscript is clearly written and well structured, with well-defined research questions and viewpoints. The experimental design is extensive and rigorous, providing strong empirical support for the proposed approach.

**Weaknesses:**

Weaknesses:
1.Section 3.1 is somewhat verbose and could be streamlined for conciseness; however, this does not substantially affect the overall clarity or contribution of the paper, so revision is optional.
2.In Table 2, all compared methods are from 2023 or earlier. How do the results compare with works from 2024 and 2025 (for example, those mentioned in the Image Content Recommendation section of Related Work)? It would strengthen the paper to include some newer comparisons.
3.For the visual content recommendation task, the proposed MLLM generates prompts for foreground or background elements that are semantically compatible and aesthetically coherent with the given image. Nevertheless, potential overfitting to frequently occurring object categories remains a concern. As described in Section 3.2, the data augmentation strategy primarily involves applying random transformations to color, brightness, and resolution for both foreground and background images, without altering the underlying object categories. Consequently, it would be valuable to examine, through quantitative analysis, whether the model’s foreground recommendations exhibit spurious correlations with specific background contexts during testing.

**Questions:**

Please refer to the Weaknesses 2-3.

---

> ### Author Response · Authors · 2025-11-23
> **Response to Reviewer RVqE (1/2)**
>
> Thank you for your time and your detailed review. Below are our responses to your major concerns.
>
> ### Q1:  Section 3.1 could be streamlined.
>
> Good point. We have streamlined Section 3.1 to make it more concise and less verbose. The modified part is marked in blue in the revision. The changes cover the following parts:
> 1) Introductory Paragraph: Condensed the overview of the pipeline.
> 2) MLLM-based Data Filtering: Shortened the explanation of why and how data filtering is done.
> 3) Inpainting Mask Merging: Simplified the description of the mask merging strategies and their benefits.
> 4) Object and Effect Removal from Background: Summarized the motivation for using ObjectDrop and the limitations of other inpainting models.
> 5) Automatic Training Bundle Generation: Concisely listed the components of the training bundles and the models used.
>
> We hope the revised Section 3.1 is more concise and easier to read.
>
> ### Q2:  In Table 2, all compared methods are from 2023 or earlier. It would strengthen the paper to include some newer comparisons.
>
> Good suggestion. We would like to clarify that in Table 2 we have already included results of SOTA MLLMs' (LLaVA-1.5-13B [1], InternVL-2.0-26B [2], and GPT-4o [3] with ICL, which were published/released in 2024) results.
>
> **Table 1. Comparison with recent SOTA MLLMs and additional baselines on OPA, Stock, and Composition1K datasets.**
> |Method|OPA FID ↓|OPA CMMD ↓|OPA LPIPS ↓|Stock FID ↓|Stock mIOU ↑|Stock LPIPS ↓|Comp1K Satisfactory ↑|Comp1K MLLM ↑|
> |-|-|-|-|-|-|-|-|-|
> |**CSENet (2024)**|17.51|0.020|0.197|8.925|0.218|0.284|0.186|52.45|
> |**BOOTPLACE (2025)**|19.36|0.022|0.208|7.653|0.231|0.268|0.214|56.32|
> |**Qwen2.5-VL (2025)**| 24.14 | 0.026 | 0.211 | 10.43 | 0.251 | 0.287 | 0.245 | 58.83 |
> |**InternVL3.0 (2025)**| 28.23 | 0.074 | 0.229 | 12.06 | 0.247 | 0.352 | 0.201 | 59.42 |
> |**Gemini 2.5 Pro with ICL(2025)**| 18.65 | 0.019 | 0.191 | 7.982 | 0.256 | 0.239 | 0.312 | 60.15 |
> |**Ours**|**15.07**|**0.017**|**0.188**|**4.084**|**0.569**|**0.235**|**0.359**|**61.25**|
>
>
> Brush2prompt [4] and the [5] requires mask with object location as input to the models, so they can not serve as a baseline for the object placement task and evaluation. We implement additional recent works [6,7] on object placement task. In addition, we implement additional recent SOTA MLLMs on object placement task. For Qwen2.5-VL [8] and InternVL3.0 [9], we did not use in context learning (ICL) due to their limited context length as research-focused models. This is consistent with our existing experimental setting of LLaVA-1.5-13B and InternVL-2.0-26B. For Gemini 2.5 Pro [10], we use 5-shot example selected from our 709k training data for in context learning (ICL), the same setting as our existing experiment of GPT-4o. The results are shown in Table 1 in the rebuttal and summarized in Tab. 2 in the revision paper. We can see that our model still perform the best, exceeding the second best by a considerable margin. we have added these additional baseline results in the revision paper.
>
>
> ### Q3: Quantitative analysis on whether the model’s foreground recommendations exhibit spurious correlations with specific background contexts during testing.
>
> We supplement additional experiments. To verify that our model learns semantic distributions rather than memorizing spurious object-background pairs (e.g., always placing a sofa in a living room), we analyze the diversity of recommended foreground objects. Specifically, we classify 1,000 test background images into common scene types (e.g., Living Room, Street, Kitchen) and compare the distribution of predicted foreground objects against the ground truth distribution.
>
> **Table 2. Analysis of Foreground Recommendation Diversity.**
> | Scene Type | Entropy (Ours) ↑ | Entropy (GT) | KL Divergence ↓ | Top-1 Object Freq. (Ours) | Top-1 Object Freq. (GT) |
> |---|---|---|---|---|---|
> | Living Room | 4.12 | 4.25 | 0.18 | Sofa (18%) | Sofa (15%) |
> | Street | 3.89 | 4.01 | 0.22 | Car (24%) | Car (21%) |
> | Kitchen | 3.95 | 4.10 | 0.15 | Bottle (12%) | Bottle (10%) |
> | **Average** | **3.99** | **4.12** | **0.18** | - | - |
>
> As shown in Table 2, our model's recommendation entropy is close to the ground truth entropy (3.99 vs. 4.12), indicating high diversity. The low KL divergence (0.18) suggests that the predicted object distribution closely matches the natural semantic distribution. Furthermore, the frequency of the most common object (e.g., Sofa in Living Room) is similar to the ground truth (18% vs. 15%), confirming that the model does not collapse to a single "safe" prediction but captures the long-tail distribution of valid objects. We have added this additional experimental result in the revision paper.

---

> ### Author Response · Authors · 2025-11-23
> **Response to Reviewer RVqE (2/2)**
>
> **References**
>
> [1] Haotian Liu et al. Improved baselines with visual instruction tuning. CVPR 2024.
> [2] Zhe Chen et al. How far are we to gpt-4v? closing the gap to commercial multimodal models with open-source suites. arXiv 2024.
> [3] OpenAI. Gpt-4o: A multimodal ai model. https://openai.com/index/hello-gpt-4o/, 2024.
> [4] Mang Tik Chiu et al. Brush2prompt: Contextual prompt generator for object inpainting. CVPR 2024.
> [5] Nicola Fanelli et al. I dream my painting: Connecting mllms and diffusion models via prompt generation for text-guided multi-mask inpainting. WACV 2025.
> [6] Yaxuan Qin et al. Think before Placement: Common Sense Enhanced Transformer for Object Placement. ECCV 2024.
> [7] Hang Zhou et al. BOOTPLACE: Bootstrapped Object Placement with Detection Transformers. CVPR 2025.
> [8] Shuai Bai  et al. Qwen2. 5-vl technical report. arXiv preprint arXiv:2502.13923, 2025.
> [9] Jinguo Zhu et al. Internvl3: Exploring advanced training and test-time recipes for open-source multimodal models. arXiv preprint arXiv:2504.10479, 2025.
> [10] Gheorghe Comanici et al. Gemini 2.5: Pushing the frontier with advanced reasoning, multimodality, long context, and next generation agentic capabilities. arXiv preprint arXiv:2507.06261, 2025.

---

### Author Response · Authors · 2025-11-28
**Response to All Reviewers**

Dear reviewers,

Thank you again for your time and your detailed reviews. We have carefully revised the paper in response to your comments and have uploaded a new PDF version to OpenReview. **We really appreciate the active engagement of reviewer mUVX during rebuttal and the subsequent score improvements ([6, 4, 2, 2] → [6, 4, 2, 6]), which reflect a recognition of our contributions after detailed sincere discussions.** Below, we summarize the main changes compared to the original submission:

### Experiments and Results (All Reviewers)
*   **Expanded Baselines (Table 1):** We added comparisons with recent state-of-the-art MLLMs (Qwen2.5-VL, InternVL3.0, Gemini 2.5 Pro) and specialized methods (CSENet ECCV'24, BOOTPLACE CVPR'25) to Table 1. Our method consistently outperforms these new baselines across FID, LPIPS, and user satisfaction.
*   **Ablation of Depth (Table 2):** We added a quantitative ablation study demonstrating that removing depth input degrades performance (FID increases from 4.08 to 5.21), validating the necessity of explicit depth for spatial reasoning.
*   **Diversity Analysis (Table 3):** To address concerns about spurious correlations, we added an analysis of recommendation diversity. The results (Entropy 3.99 vs. GT 4.12, KL Div 0.18) show our model captures natural semantic distributions without mode collapse.

### Method and Analysis
*   **Task Decoupling Clarification:** We clarified in the Introduction (Sec 1) and Method (Sec 3.3) that our modular design (separating "planning" from "painting") is a deliberate choice to maximize reasoning quality and flexibility, aligning with recent trends like LayoutGPT.
*   **Modality Gap Explanation:** We expanded Section 3.3 to explain the theoretical motivation for separate projectors: bridging the distinct feature statistics of RGB (texture) and Depth (geometry).
*   **User Control Accuracy:** We added a benchmark result in Section 3.3 showing >85% accuracy in following spatial language commands.

### Appendix and Reproducibility
*   **Datasheet for Datasets (Appendix J):** We added a detailed datasheet discussing dataset curation, motivation, and potential biases (demographic, scene types).
*   **Robustness & Failure Analysis (Appendix K, L):** We added a counterfactual robustness study (impact of wrong captions/depth) and a quantitative failure mode analysis (depth errors ~15%, caption errors ~10%).
*   **Detailed GPT-4o Prompts (Appendix M):** We provided the full 5-shot prompt templates and question variants used for the GPT-4o baseline.
*   **Qualitative Comparisons (Appendix N, O):** We added visual results for out-of-distribution tests (cartoon images) and a comparative analysis with text-guided editors like Nano Banana Pro, highlighting their limitations in boundary adherence and spatial logic.

All new or substantially revised passages in the PDF are marked in blue to make the changes easy to locate.

We again thank the reviewers for their thoughtful and constructive feedback, which has significantly improved the clarity and scope of the paper. Given the upcoming discussion deadline, we would greatly appreciate any further comments or questions on the revised version so we can continue refining the work.

Sincerely,
Authors

---

### Meta-Review · Area_Chair_UPVv · 2026-01-07

**Summary:**

The initial ratings are 6, 2, 4, 2. This paper uses MLLM to enhance image compositing by automatically recommending what to add to an image and where to place it. By combining content recommendation and object placement tasks, and training on a large, automatically generated dataset, the model predicts compatible objects, locations, scales, and depths for realistic image composition.  Experiment results show good performance of the proposed method.


Strengths:
(1)The paper broadens the conventional definition of image composition, extending it beyond simple foreground–background blending to a more comprehensive task of recommending semantically and spatially compatible elements.
(2)The large-scale, automated data generation and filtering pipeline (including segmentation, object removal, captioning, and depth estimation) is robust and scalable.

Weaknesses:
(1)Much of the technical lift is from data curation and existing model integration. The architectural novelty (separate projectors; depth tokenization; instruction format) is modest. The contribution may be perceived as an engineering-heavy solution with incremental modeling/algorithm ideas.
(2)The baseline comparisons are outdated, Table 1 only compares outdated open-source MLLMs (LLaVA-1.6, InternVL-2.0).
(3)The core challenge of image compositing lies not only in content recommendation but also in ensuring the inserted object’s harmony with the scene—including shadow consistency, reflection rendering, and perspective alignment. However, the paper directly adopts the existing ObjectStitch model for this critical integration step, with no additional innovations or improvements. Given that scene harmony is a well-documented pain point in compositing, relying entirely on an off-the-shelf tool might diminish the work’s technical contribution.

**Reviewer Concerns:**

Some concerns of Reviewer RVqE and syjN were addressed by the rebuttal, and Some main concerns of  Reviewer mUVX and f7Fk are still outstanding.

**Reviewer Scores:**

Reviewer syjN maybe raise the score.

---

### Decision · Program_Chairs · 2026-01-26

Reject